# A Robust Optimization Guided Pruning Framework for Vision and Large Language Models

Gabriel Afriat [1] [†]   Hussein Hazimeh [2] [*]   Dimitris Paparas [3]   Rahul Mazumder [1] [4]

## Abstract

Pruning is a common approach to reduce the memory footprint and inference cost of large vision and language models. As these architectures continue to scale, one-shot pruning methods - i.e. approaches that prune the network without any retraining - have become increasingly attractive. Many popular one-shot pruning methods (e.g., WoodFisher, CAP, SparseGPT, and ALPS) typically optimize a quadratic objective under sparsity constraints. However, in practice, this objective is affected by multiple sources of uncertainty, including noise in the calibration data and variability introduced by algorithmic updates. To address these issues, we introduce *RobOP*, a robust optimization framework that explicitly accounts for such uncertainties. *RobOP* is modular and flexible, and can be applied with any existing pruning method through simple modifications motivated by our theoretical framework. We demonstrate that by taking into account uncertainty, *RobOP* offers improvements over prior pruning approaches. Our framework applies tractably across a range of stylized uncertainty sets, enabling robust one-shot pruning at scale. Our code is available at https://github.com/mazumder-lab/RobOP.

## 1. Introduction

Recent advances in computer vision and Large Language Models (LLMs) have achieved strong empirical performance, but often at the cost of steadily increasing parameter counts and compute requirements (He et al., 2016; Dosovitskiy et al., 2021; Zhang et al., 2022; Touvron et al., 2023a;b; Grattafiori et al., 2024). This trend makes deployment increasingly costly, especially when serving billion-parameter LLMs at scale, and limits the use of high-capacity vision models on resource-constrained edge devices. These computational and deployment challenges motivate model compression methods that reduce inference costs while preserving accuracy. Among these, pruning is a simple yet effective approach that introduces sparsity by setting weights to zero. In practice, pruning can reduce memory footprint and significantly accelerate inference (Kuznedelev et al., 2023; Frantar & Alistarh, 2023; Sun et al., 2024; Benbaki et al., 2023; Meng et al., 2024a).

**Pruning without Retraining.** Many methods have been proposed for pruning pre-trained vision models and LLMs (Han et al., 2015; Frankle & Carbin, 2019; Hoefler et al., 2021; Yu et al., 2022; Frantar & Alistarh, 2022; Benbaki et al., 2023; Kuznedelev et al., 2023; Frantar & Alistarh, 2023; Meng et al., 2024b; Sun et al., 2024; Meng et al., 2024a). While some of these approaches rely on retraining the sparse subnetwork to recover performance (Han et al., 2015; Frankle & Carbin, 2019), this can be very expensive for modern large-scale architectures. As a result, recent work (Kuznedelev et al., 2023; Frantar & Alistarh, 2023; Meng et al., 2024a) has focused on ways to prune the network in one-shot, i.e. without any retraining.

**Pruning Objectives.** In pruning, a function of the model weights (a proxy for model utility) is typically considered to decide which weights to set to zero. For example, in unstructured pruning, the magnitude of the weights can be used to determine their impact on model utility (Hanson & Pratt, 1988; Mozer & Smolensky, 1989; Gordon et al., 2020). While simple and computationally efficient, magnitude pruning typically incurs a substantial loss in accuracy at moderate sparsity levels and often requires expensive model retraining to recover performance. To better preserve model performance without retraining, state-of-the-art one-shot pruning methods for vision models (Benbaki et al., 2023; Kuznedelev et al., 2023) leverage a second-order Taylor approximation of the training loss: $\mathcal{L}(\bar{w} + d) \approx \mathcal{L}(\bar{w}) + \nabla\mathcal{L}(\bar{w})^\top d + \frac{1}{2}d^\top \nabla^2\mathcal{L}(\bar{w})d$ and assume $\nabla\mathcal{L}(\bar{w}) = 0$ given that the network is pre-trained. Here $d$ denotes the weight update that transforms the dense

[†]Part of this work was done during an internship at Google. [*]Work done while at Google. [1]Operations Research Center, MIT, Cambridge, USA [2]OpenAI, San Francisco, CA, USA [3]Google Research, Mountain View, CA, USA [4]Sloan School of Management and Center for Statistics, MIT, Cambridge, USA. Correspondence to: Gabriel Afriat <afriatg@mit.edu>.

*Proceedings of the 43rd International Conference on Machine Learning*, Seoul, South Korea. PMLR 306, 2026. Copyright 2026 by the author(s).

weights $\bar{w}$ into the pruned weights $w = \bar{w} + d$. These methods consider the impact of pruning on the training loss:

$$\mathcal{L}_F(w, \bar{w}) := \mathcal{L}(\bar{w} + d) - \mathcal{L}(\bar{w}) \approx \frac{1}{2} d^\top \nabla^2 \mathcal{L}(\bar{w}) d \quad (1)$$

In practice, the Hessian $\nabla^2 \mathcal{L}(\bar{w})$ is approximated by the empirical Fisher matrix: $H_F = \frac{1}{N} \sum_{i=1}^{N} \nabla \ell_i(\bar{w}) \nabla \ell_i(\bar{w})^\top$, where $\nabla \ell_i(\bar{w})$ is the gradient of the loss with respect to the weights $\bar{w}$ on the $i$-th training sample. For scalability, the positive semi-definite (PSD) matrix $H_F$ is usually approximated as a block-diagonal matrix: $H_F = \text{Diag}(H_1, \ldots, H_K)$. The resulting proxy objective is then minimized to choose weights for pruning and to update the support until the target sparsity budget is reached. For LLMs, a line of recent work (Frantar & Alistarh, 2023; Sun et al., 2024; Meng et al., 2024a) optimizes, subject to a sparsity budget on the weights, a layer-wise reconstruction criterion, typically of the form:

$$\mathcal{L}_R(W, \bar{W}) = \|\bar{W}X - WX\|_F^2 \quad (2)$$

where $\bar{W}, W \in \mathbb{R}^{d_{\text{in}} \times d_{\text{out}}}$ denote the dense and pruned layer weights respectively and $X \in \mathbb{R}^{N \times d_{\text{in}}}$ the layer input. Here also we can write the loss in (2) as a function of the weight update $d = w - \bar{w}$:

$$\mathcal{L}_R(W, \bar{W}) = (w - \bar{w})^\top H_R (w - \bar{w}) \quad (3)$$

with $w$ and $\bar{w}$ the vector form of $W$ and $\bar{W}$ respectively, and PSD matrix $H_R = \text{Diag}(XX^\top, \ldots, XX^\top)$ where $XX^\top$ is repeated $d_{\text{out}}$ times. These methods prune (and update) the weights to minimally impact the difference between the outputs of the pruned and dense layers. Despite considering different criteria, both the layer-wise reconstruction and Fisher approximation approaches optimize a convex quadratic objective subject to sparsity constraints to prune and update the weights:

$$\min_d \ \frac{1}{2} d^\top H d \ \ s.t. \ \ \bar{w} + d \in \mathcal{S} \quad (4)$$

with $H = H_F$ (Fisher) or $H = H_R$ (layer-wise). Here, the constraint $\bar{w} + d \in \mathcal{S}$ enforces the resulting weights to belong to set $\mathcal{S}$, which encodes the desired sparsity properties (e.g., maximum number of non-zero entries, sparsity pattern, or other structural constraints).

**Finite-Sample Noise in Hessian Estimation.** Formulation (4) requires an estimate of the Hessian matrix $H$. In the case of Fisher-based approaches, $H_F$ is computed using only a limited calibration set–typically a few thousand samples (Singh & Alistarh, 2020a; Benbaki et al., 2023; Kuznedelev et al., 2023) even for pruning large CNNs and Vision Transformers (ViTs). Similarly, layer-wise reconstruction criteria are usually evaluated on a small calibration set. For example, SparseGPT (Frantar & Alistarh, 2023)

and ALPS (Meng et al., 2024a) use 128 sequences of 2048 tokens each to compute $H_R$. Being able to prune using a small calibration set is often desirable in practice, as it enables post-training compression on downstream tasks with limited data and at lower computational cost. At the same time, these practices raise a central question: *Can one-shot pruning be made robust to uncertainty arising from the use of small and finite calibration sets?*

**Hessian Uncertainty from Algorithmic Updates.** For the layer-wise reconstruction loss, the pruning criterion for a given layer is computed using the layer inputs $X$, obtained by passing the calibration data through all preceding layers. Since leading pruning methods proceed sequentially over the decoder blocks, earlier blocks (and thus the layers they contain) have already been sparsified. Consequently, the Hessian estimate used for the current layer reflects the network state after pruning the earlier blocks. On the other hand, Fisher-based approaches typically compute gradients (and the empirical Fisher matrix $H_F$ for each layer) only once using the dense network, and then reuse these estimates to score and prune weights across the entire network. This creates a mismatch: once earlier layers are pruned, both the gradients and the Hessian will change, so the original estimates obtained from the dense model may no longer serve as a good estimate for the partially pruned network. This leads to a second key question: *Can we make Fisher-based pruning methods robust to Hessian estimates computed once on the dense model and not updated during the pruning process?*

**Robust Optimization for One-Shot Pruning.** Motivated by the above discussion, here we propose a new approach to manage various uncertainties arising in the pipeline of two widely used pruning approaches: (i) Fisher-based approaches for vision models and (ii) layer-wise reconstruction formulations for LLMs. Specifically, we focus on how to manage uncertainty in estimating $H$ as they arise from sampling errors, modeling approximations (e.g., Fisher approximation in place of the true Hessian), and changes in the gradients and Hessians during the algorithm updates. To this end, we propose *RobOP* (**Rob**ust **O**ptimization for one-shot **P**runing), a new robust optimization framework that can mitigate these uncertainties. Robust optimization is a principled approach in mathematical optimization for handling uncertainty in optimization problems (Bertsimas et al., 2011). Using robust optimization principles (Bertsimas et al., 2011), *RobOP* replaces the standard (or, nominal) quadratic objective in (4) used by popular baselines with a robust variant that incorporates uncertainties in estimating $H$. Under this robust optimization setting, we consider uncertainty sets that capture different sources of error in estimating $H$. They result in optimization formulations that preserve the same structure and can be viewed as variants of (4). Indeed, a key observation is that these robust

formulations are given by simple modifications to the original non-robust formulations where we add a simple convex quadratic regularization term to the objective which can be efficiently computed. As the uncertainty aware robust optimization problems have a structure similar to the original non-robust problems, they can be optimized with existing pruning methods with minimal algorithmic changes and overhead — scaling to modern vision and LLMs with millions to billions of parameters.

**Contributions.** Our contributions are as follows:

- We present *RobOP*, a robust optimization framework for one-shot pruning. *RobOP* takes into account uncertainties in estimating the Hessian matrix arising in popular one-shot pruning approaches. We derive computationally appealing reformulations of these robust optimization problems for different choices of the uncertainty set; and demonstrate how existing pruning approaches can be modified to address these problems.

- We evaluate the impact of *RobOP* on Vision Transformers by adapting CAP (Kuznedelev et al., 2023) and pruning the DeiT model family (Touvron et al., 2021). We show that the robust formulations consistently improve accuracy, yielding gains of up to 10 points in test accuracy on ImageNet-1K (Deng et al., 2009).

- We evaluate the impact of *RobOP* on LLMs by adapting ALPS (Meng et al., 2024a) and pruning Llama-3.1-8B (Grattafiori et al., 2024) and Llama-2-13b-hf (Touvron et al., 2023b). We find that robust formulations provide up to 10-point improvements in downstream WikiText2 perplexity and a 0.7-point gain in mean accuracy on 3 common reasoning classification tasks.

## 2. Related Work

**Neural Network Pruning**: Pruning is a well-known approach to reduce the complexity of large neural networks (Hassibi & Stork, 1992b; Han et al., 2015; Singh & Alistarh, 2020a). Pruning can be (a) unstructured, where any weight can be set to 0 (Han et al., 2015; Singh & Alistarh, 2020a; Kuznedelev et al., 2023; Frantar & Alistarh, 2022; 2023; Meng et al., 2024a), (b) structured, where entire rows or columns are set to 0 (Ma et al., 2023; Meng et al., 2024c), or (c) semi-structured where $n$ weights have to be set to 0 for any block of $m$ weights (Singh & Alistarh, 2020a; Kuznedelev et al., 2023; Frantar & Alistarh, 2022; 2023). Unstructured pruning typically achieves the best sparsity-accuracy trade-off, and can yield substantial memory savings and speedups on CPUs (NeuralMagic, 2021; Frantar & Alistarh, 2023) and specialized hardware accelerators (Han et al., 2015; Dave et al., 2021). Semi-structured and structured pruning are usually used at lower sparsity levels

but offer speedups on GPUs (Frantar & Alistarh, 2023; Sun et al., 2024; Meng et al., 2024c). In this work, we focus on unstructured and semi-structured pruning.

**One-shot Pruning**: In order to limit performance degradation during pruning, some methods propose to retrain the network at the end of pruning, or to perform gradual pruning where they alternate phases of pruning with phases of retraining (Han et al., 2015; Gale et al., 2019; Singh & Alistarh, 2020a; Blalock et al., 2020; Benbaki et al., 2023). Other methods propose to prune the network during training or at initialization (Louizos et al., 2017; Frankle & Carbin, 2019; Lee et al., 2019; Liu et al., 2019; Lee et al., 2020; Wang et al., 2020; Renda et al., 2020; Frankle et al., 2021; Zhang et al., 2021). While effective, these methods require expensive retraining and are often too costly for recent large vision and language models. To address this issue, recent research has focused on pruning weights of pre-trained networks in a one-shot manner, i.e. without any retraining.

**Pruning Vision Models**: For vision models, a popular family of one-shot pruning methods is based on the OBD and OBS (Optimal Brain Damage/Surgeon) framework (LeCun et al., 1989; Hassibi & Stork, 1992a). These methods (see Section 1) consider a quadratic loss function using the empirical Fisher matrix to approximate the Hessian. In order to scale the OBS framework to more modern, million-parameter scaled, architectures, Singh & Alistarh (2020b) propose a block-diagonal approximation of the empirical Fisher matrix. Later works explore pruning based on joint weight interactions (Yu et al., 2022) or reformulate the objective to exploit the Fisher structure, optimizing it with iterative thresholding (Benbaki et al., 2023). More recently, Lucas & Mazumder (2025) propose to optimize a global reconstruction objective with Hessian-free second-order optimization. However, they focus on refining an existing pruning mask. Kuznedelev et al. (2023), with their Correlation Aware Pruner (CAP), adapt the greedy OBS algorithm to prune large ViTs from scratch. They use a block-diagonal approximation of $H$ and apply rank-1 updates to $H^{-1}$ to account for weight removal. Their algorithm achieves state-of-the-art performance on ViTs.

**Pruning LLMs**: For LLMs, state-of-the-art one-shot pruning approaches use a layer-wise reconstruction error formulation (Frantar & Alistarh, 2023; Sun et al., 2024; Meng et al., 2024a) (See Section 1). This pruning objective is also a convex quadratic, like the Fisher-based objectives. Thus, methods such as OBC by Frantar & Alistarh (2022) propose to adapt the OBS framework to this new loss function and take advantage of the exact block-diagonal structure of $H_R$. To scale to billion parameter LLMs, Frantar & Alistarh (2023) prune the weight matrix by blocks of columns and use this strategy to reduce the computational cost for subsequent blocks by considering the Hessian for the remaining

columns only. These approaches use greedy heuristics to optimize the problem in (4) inspired by OBS(Hassibi & Stork, 1992b). Meng et al. (2024a) introduce ALPS which uses ADMM (and preconditioned conjugate gradient) to simultaneously find the support and update the weights using a layer-wise loss function. ALPS shows state-of-the-art performance for one-shot unstructured pruning of LLMs.

**Robust Pruning**: While there are prior works in the literature on robust pruning, they have a different focus than ours. Prior works study finding a robust set of weights for pruning at initialization (Hayou et al., 2021), or pruning networks while preserving robustness against adversarial attacks (Zhao & Wressnegger, 2023; Sehwag et al., 2020). These approaches typically rely on retraining and focus on small CNN architectures. In contrast, rather than focusing on adversarial attacks, we propose a robust one-shot pruning method aimed at improving model performance under typical use conditions. Li et al. (2023) study robust pruning for LLMs, focusing on robustness to input-level attacks (wrong tokens). However, they focus on relatively small LLMs (BERT$_{\text{base}}$ with around 110 million parameters) and their approach relies on training multiple models with different hyperparameter settings to find robust ones. To the best of our knowledge, our work is the first to consider uncertainty in the pruning criteria (stemming from noise in the Hessian estimates – see Section 1) for one-shot pruning of ViTs and LLMs. Additionally, for different uncertainty sets, we theoretically derive robust counterparts for popular pruning formulations (1) and (2), which can be computed with existing pruning algorithms.

## 3. Robust Pruning Formulations

### 3.1. Robust Pruning: Fisher-based pruning methods

For Fisher-based approaches, uncertainty arises from three main sources: (i) approximating the Hessian $H$ with the empirical Fisher matrix, (ii) estimating $H$ from a finite number of gradients, and (iii) changes in per-sample gradients and $H$ during pruning. Without frequent recomputation, $H$ becomes outdated. However, frequent recomputation can be prohibitively expensive, which is why recent approaches such as CAP (Kuznedelev et al., 2023) compute $H$ only once. To mitigate these sources of uncertainty on $H$, we consider the following robust optimization problem:

$$\min_{d} \quad \max_{\|\Delta\|_F \leq \gamma} \quad \frac{1}{2}d^\top(H + \Delta)d \qquad (5)$$
$$s.t. \quad \bar{w} + d \in \mathcal{S}$$

**Theorem 3.1.** *Problem* (5) *is equivalent to solving*

$$\min_{d} \frac{1}{2}d^\top(H + \gamma I)d \quad s.t. \quad \bar{w} + d \in \mathcal{S} \qquad (6)$$

We note that adding a $\gamma I$ term to the Hessian is a common trick in the pruning literature which uses the Fisher approximation (Singh & Alistarh, 2020a; Kuznedelev et al., 2023; Benbaki et al., 2023). However, a key motivation in prior work appears to be numerical stability (Kuznedelev et al., 2023; Frantar & Alistarh, 2023): the OBS update requires inverting $H$, which can be rank deficient, and a dampening term $\gamma I$ can be added to make $H$ positive definite. These approaches do not use robust optimization principles and propose using a fixed and small value as default ($10^{-8}$ for CAP, $10^{-5}$ for WoodFisher).

While (5) considers a fixed uncertainty radius $\gamma$ on the Frobenius norm of $H$, we also consider more tailored uncertainty sets. In particular, given that $H$ is estimated using the empirical Fisher matrix, it can be written as $H = \frac{1}{N}AA^\top$ with $A = \begin{pmatrix} \nabla\ell_1(\bar{w}) & \dots & |\nabla\ell_N(\bar{w}) \end{pmatrix}$. Let $\epsilon = \begin{pmatrix} \epsilon_1 & \dots & \epsilon_N \end{pmatrix}^\top$ be a pre-specified vector. We propose to consider uncertainty in the per-sample gradients:

$$\min_{d} \quad \max_{\|\Delta_{:,i}\|_2 \leq \epsilon_i \ \forall i} \quad \frac{1}{2N}d^\top(A + \Delta)(A + \Delta)^\top d \quad (7)$$
$$s.t. \quad \bar{w} + d \in \mathcal{S}$$

with $\Delta_{:,i}$ the $i$-th column of $\Delta$.

**Proposition 3.2.** *Problem* (7) *is equivalent to solving*

$$\min_{d} \frac{1}{2N}\left\|\,|A^\top d| + \|d\|_2\epsilon\right\|_2^2 \quad s.t. \quad \bar{w} + d \in \mathcal{S} \quad (8)$$

In the case of CAP, this problem is solved for each weight in the support by considering $\mathcal{S} = \left\{ w \mid (\bar{w} + d)^\top e_q = 0 \right\}$ for all possible $q$. For this choice of $\mathcal{S}$, a closed-form solution exists for the nominal problem (4), but not for (8). While (8) can be solved using a second-order cone programming solver, solving it for all the weights would cause major computational overhead. Hence, for computational reasons, we consider the following problem:

**Theorem 3.3.** *Optimal objective of Problem* (8) *is upper bounded by that of the following problem:*

$$\min_{d} \ d^\top(H + \frac{1}{N}\|\epsilon\|_2^2 I)d \quad s.t. \ \bar{w} + d \in \mathcal{S} \qquad (9)$$

Note that the objective in problem (9) is again quadratic in $d$; and a solution can be computed based on simple modifications of existing one-shot pruning approaches. We note that choosing $\epsilon_i = \gamma$ a constant brings us back to the optimization problem of Theorem 3.1. However, we may assume that $\epsilon_i = \gamma\|\nabla\ell_i(\bar{w})\|_2$, i.e. the uncertainty on the per-sample gradient depends on the magnitude of $\nabla\ell_i(\bar{w})$. In this case, we find that $\frac{1}{N}\|\epsilon\|_2^2 = \frac{1}{N}\sum_{i=1}^N \gamma^2\|\nabla\ell_i(\bar{w})\|_2^2 = \gamma^2\text{Tr}(\frac{1}{N}AA^\top) = \gamma^2\text{Tr}(H)$.

**Regularization term in the block-diagonal case.** In practice, prior works (Singh & Alistarh, 2020a; Kuznedelev

et al., 2023) often approximate $H$ by a block-diagonal matrix, $H = \mathrm{Diag}(H_1, \ldots, H_K)$. Only the diagonal blocks are estimated using the Fisher approximation. Rather than modeling uncertainty in the full matrix, we model uncertainty at the block level by restricting $\Delta$ to be block-diagonal: $\Delta = \mathrm{Diag}(\Delta_1, \ldots, \Delta_K)$. The weight updates can also be defined blockwise: $d = (d_1, \ldots, d_K)^\top$, where each $d_k$ has the same dimension as $H_k$. Since the objective is separable across blocks, it follows that:

$$\max_{\Delta_k \in \mathcal{U}_k \forall k} \frac{1}{2} d^\top (H + \Delta) d = \sum_{k=1}^{K} \max_{\Delta_k \in \mathcal{U}_k} \frac{1}{2} d_k^\top (H_k + \Delta_k) d_k \tag{10}$$

with $\mathcal{U}_k$ the uncertainty set on $\Delta_k$. In light of the above decomposition, the results in Theorems 3.1 and 3.3 apply to (10) at the block level for every block subproblem in (10).

### 3.2. Robust Pruning: layer-wise reconstruction error

In the case of the layer-wise reconstruction error, state-of-the-art approaches usually prune the decoder blocks sequentially, which implies that before pruning a layer, its input $X$ is obtained from passing the input data through the previous pruned decoder blocks. $H$ is therefore largely up to date. In addition, pruning a layer doesn't affect its input data and $XX^\top$. A major source of uncertainty in this case is the finite number of samples used to estimate $H$.

Several prior works (Bunea & Xiao, 2015; Loukas, 2017; Puchkin et al., 2024) establish concentration bounds for the sample covariance matrix. However, given that the input of each layer is not necessarily centered, $\frac{1}{N} XX^\top$ can be interpreted as a sample second moment matrix. The following results focus on deriving new concentration bounds for the sample second moment matrix.

Let's denote $X_i$ as the random variable corresponding to the $i$-th sample. We consider the same setting as in Bunea & Xiao (2015) and use the same assumption (Assumption 1 in their paper, see Appendix A for the exact assumption).

**Theorem 3.4.** *Let $X_i$ be i.i.d., $X_i \sim Y$ with $Y$ sub-Gaussian. Let $M = \mathbb{E}[YY^\top]$ and $\hat{M} = \frac{1}{N} \sum_{i=1}^{N} X_i X_i^\top$. If $Y - \mathbb{E}[Y]$ satisfies Assumption A.1, then:*

$$\mathbb{E}\left[\|M - \hat{M}\|_F^2\right] \lesssim (1/N) Tr(M)^2$$

A similar theorem is proved by Bunea & Xiao (2015) for the sample covariance matrix and in the case where $\mathbb{E}[Y] = 0$. Theorem 3.4 extends the result to the sample second moment matrix without any assumption on $\mathbb{E}[Y]$.

Theorem 3.4 motivates an uncertainty set for the Hessian

and a corresponding robust optimization problem:

$$\min_{d} \max_{\|\Delta\|_F \leq \gamma \frac{Tr(XX^\top)}{\sqrt{N}}} \frac{1}{2} d^\top (H + \mathrm{Diag}(\Delta, \ldots, \Delta)) d \tag{11}$$

$$s.t. \quad \bar{w} + d \in \mathcal{S}$$

Indeed, $XX^\top$ as defined earlier corresponds to the realization of the random variable $\hat{M}$. We expect $XX^\top$ to be close to the second order moment matrix $M$, by an amount $\frac{Tr(XX^\top)}{\sqrt{N}}$ (see Theorem 3.4).

**Corollary 3.5.** *Problem* (11) *is equivalent to solving*

$$\min_{d} \frac{1}{2} d^\top \left(H + \gamma \frac{Tr(XX^\top)}{\sqrt{N}} I\right) d \tag{12}$$

$$s.t. \quad \bar{w} + d \in \mathcal{S}.$$

This is a direct consequence of Theorem 3.1 replacing $\gamma$ by $\gamma \frac{Tr(XX^\top)}{\sqrt{N}}$.

While Theorem 3.5 proposes to bound the Frobenius norm of $\Delta$, other concentration bounds can be derived on the sample second moment matrix $\hat{M}$. Loukas (2017) proposes concentration bounds to measure the uncertainty on the eigenvalues of the sample covariance matrix. We observe that this theorem holds also in the case of the sample second moment matrix $\hat{M}$:

**Theorem 3.6.** *(Loukas (2017)) Let $X_i$ be i.i.d., $X_i \sim Y$. Let's define $M = \mathbb{E}[YY^\top]$ and $\hat{M} = \frac{1}{N} \sum_{i=1}^{N} X_i X_i^\top$. Let's note $\lambda_1, \ldots, \lambda_p$ the eigenvalues of $M$ and $\hat{\lambda}_1, \ldots, \hat{\lambda}_p$ the eigenvalues of $\hat{M}$. Let's also note $u_1, \ldots, u_p$ the eigenvectors of $M$. We have that with probability at least $1 - \delta$*

$$|\hat{\lambda}_i - \lambda_i| \leq \frac{\kappa_i}{\sqrt{\delta N}} \tag{13}$$

*with $\kappa_i = \sqrt{\mathbb{E}[\|YY^\top u_i\|_2^2 - \lambda_i^2]}$.*

**Lemma 3.7.** *We consider the same setup as Theorem 3.6 and we assume that $Y$ is sub-Gaussian and that $Y - \mathbb{E}[Y]$ verifies assumption A.1, then for all $i$,*

$$\kappa_i \lesssim \sqrt{\lambda_i Tr(M)} \tag{14}$$

We note in the following $D(x) = \mathrm{Diag}(x_1, \ldots, x_{d_{in}})$ and $F(x) = \mathrm{Diag}(D(x), \ldots, D(x))$ (repeated $d_{out}$ times) for all $x \in \mathbb{R}^{d_{in}}$.

Let's consider the eigenvalue decomposition of $XX^\top$ which is positive semi-definite: $XX^\top = PD(\xi)P^\top$ with $\xi = (\xi_1, \ldots, \xi_{d_{in}})^\top$ the eigenvalues of $XX^\top$. In the layer-wise reconstruction case, $H = H_R = \mathrm{Diag}(XX^\top, \ldots, XX^\top)$. Therefore, we can write $H = U\mathrm{Diag}(D(\xi), \ldots, D(\xi))U^\top$ with $U = \mathrm{Diag}(P, \ldots, P)$.

Assuming that the eigenvectors of $\hat{M}$ are close to those of $M$, we model uncertainty at the eigenvalue level and consider the following uncertainty set:

$$\min_d \max_{|\mu_i| \leq \gamma\sqrt{\frac{\xi_i \text{Tr}(XX^\top)}{N}}} \frac{1}{2} d^\top (H + UF(\mu)U^\top)d \quad (15)$$
$$s.t. \quad \bar{w} + d \in \mathcal{S}$$

**Theorem 3.8.** *Problem* (15) *is equivalent to solving*

$$\min_d \frac{1}{2} d^\top (H + \gamma\sqrt{\frac{Tr(XX^\top)}{N}} UF(\sqrt{\xi})U^\top)d \quad (16)$$
$$s.t. \quad \bar{w} + d \in \mathcal{S}$$

**Extension to the Empirical Fisher Matrix.** While the two previous uncertainty sets are motivated using concentration bounds on the sample second-moment matrix for the layer-wise reconstruction error, the same argument extends directly to the empirical Fisher matrix: $H = \frac{1}{N} \sum_{i=1}^N \nabla \ell_i(\bar{w})\nabla \ell_i(\bar{w})^\top$. Indeed, we can similarly view $\nabla \ell_i(\bar{w})$ as the realization of a random variable $Y$, and uncertainty in $H$ arises from the finite number of samples used. In the case of the eigenvalue-wise uncertainty, and for both the layer-wise reconstruction and Fisher loss, we can note $\eta$ the eigenvalues of $H$ and the regularization term becomes:

$$R(\gamma, H) = \gamma\sqrt{\frac{\text{Tr}(H)}{N}} UD(\sqrt{\eta})U^\top \quad (17)$$

**Integrating *RobOP* into existing pruning frameworks.** Theorems 3.1, 3.3, 3.5, and 3.8 show that uncertainty in the pruning criterion can be incorporated by adding a regularization term to the Hessian $H$. As a result, *RobOP* can robustify existing pruning algorithms by changing only the computation of H, requiring minimal implementation overhead.

**Theorem assumptions in the case of ViTs/LLMs.** The assumptions made here are standard covariance concentration assumptions (Koltchinskii & Lounici, 2014; Bunea & Xiao, 2015; Loukas, 2017). While the i.i.d. assumption can be restrictive for LLMs (due to within-sentence token dependencies), it is reasonable for ViTs, where input samples are selected at random. The sub-Gaussian assumption is also appropriate in this setting: ViT inputs are bounded by the pixel range, and LLM inputs by the embedding weights. After finitely many layers, the resulting layer inputs remain bounded and hence sub-Gaussian.

### 3.3. Interpreting the dampening terms used in the literature

We note that similar regularization terms—notably $\gamma I$ for Fisher and $\gamma \frac{\text{Tr}(XX^\top)}{p} I$ for layer-wise approaches, where $p$

denotes the dimension of $XX^\top$—have appeared in prior pruning work (Singh & Alistarh, 2020a; Kuznedelev et al., 2023; Frantar & Alistarh, 2023; Meng et al., 2024a). In these works, the proposed pruning procedures (4) require inverting $H$. These regularization terms are referred to as "dampening" and primarily motivated for numerical stability when inverting $H$ (Kuznedelev et al., 2023; Frantar & Alistarh, 2023). Our work provides a principled justification for these algorithmic choices, unifies them under a general robust optimization framework, and leverages the resulting formulations to improve pruning performance. This robust optimization perspective, and its connection to one-shot pruning, is novel to our understanding.

## 4. Experimental Results

### 4.1. Models, Baselines and Datasets

We test the impact of robust pruning both in the case of ViTs and LLMs. In particular, we prune:

- The DeiT Vision Transformers (Touvron et al., 2021) using CAP (Kuznedelev et al., 2023)

- Llama-2-13b-hf (Touvron et al., 2023b) and Llama3.1-8B (Grattafiori et al., 2024) using ALPS (Meng et al., 2024a)

For pruning, we use 128 samples selected at random from the C4 dataset (Raffel et al., 2020) for the LLM experiments, and 4096 samples from ImageNet-1K (Deng et al., 2009) using a random stratified split for the ViTs. For the DeiT models, we report the test accuracy on ImageNet-1K (Deng et al., 2009). For the Llama models, we evaluate both language quality in terms of perplexity and downstream performance on classification tasks. Following prior work (Frantar & Alistarh, 2023; Sun et al., 2024), we measure test perplexity on C4, WikiText2 (Merity et al., 2016) and PTB (Marcus et al., 1994), and we additionally measure zero-shot accuracy on standard commonsense reasoning benchmarks (OpenBookQA (Mihaylov et al., 2018), WinoGrande (Sakaguchi et al., 2021), and PIQA (Bisk et al., 2020)).

ALPS focuses on unstructured pruning, so we evaluate Llama models at 60%, 70%, and 80% uniform unstructured sparsity. CAP supports both unstructured and semi-structured sparsity, so we prune DeiT models at 50%, 60%, 70%, 80% uniform unstructured sparsity, and 2:4 sparsity.

### 4.2. Implementation Details

**Setup.** We prune the DeiT models and Llama-3.1-8B using a single NVIDIA L40 GPU with 40GB of RAM. For Llama-2-13b-hf, we use an A100 with 80GB of RAM.

**Regularization terms.** For both the vision models and

LLMs, we consider the following regularization terms:

- $\gamma I$: This corresponds to $\|\Delta\|_F \leq \gamma$ (problem (5)), or in the Fisher case, to $\|\Delta_{:,i}\|_2 \leq \sqrt{\gamma}$ (problem (7)).

- $\gamma \frac{\mathrm{Tr}(H)}{\sqrt{N}}$: This corresponds to $\|\Delta\|_F \leq \gamma \frac{\mathrm{Tr}(H)}{\sqrt{N}}$ (problem (11)), or in the Fisher case, to $\|\Delta_{:,i}\|_2 \leq \sqrt{\gamma} \frac{\|\nabla \ell_i(\bar{w})\|_2}{N^{1/4}}$ (problem (7)).

- $R(\gamma, H)$: This corresponds to $|\mu_i| \leq \gamma \sqrt{\frac{\eta_i \mathrm{Tr}(H)}{N}}$ (problem (15)).

While we write $H$ in the uncertainty sets and resulting regularizers for notational simplicity, these terms are, as noted at the end of Section 3.1, added at the block level.

**Tuning of uncertainty size.** We tune $\gamma$ over the grid:

$$\gamma \in \left\{ 10^{-7}, \ldots, 5 \times 10^{-1}, 1, 5 \right\}.$$

For ViTs, we additionally try $\gamma \in \{10^{-8}, 5 \times 10^{-8}, 10, 50\}$.

To select the best value of $\gamma$, we use an additional validation set of 4096 ImageNet-1K samples for the ViTs and 1024 C4 samples for the LLMs, and choose $\gamma$ to minimize the corresponding validation loss or validation perplexity.

**Regularization term for the Baselines.** CAP (Kuznedelev et al., 2023) and ALPS (Meng et al., 2024a) use a regularization term as well in their algorithm. However, their motivation is solely numerical stability to make $H$ invertible.

- CAP (Kuznedelev et al., 2023) adds $\gamma I$ to $H$. However, they choose $\gamma$ very small and fix $\gamma = 10^{-8}$.

- ALPS (Meng et al., 2024a) adds $\gamma \frac{\mathrm{Tr}(XX^\top)}{p} I$ to $H$. However, they again fix $\gamma = 10^{-2}$.

**Numerical precision.** To mitigate numerical instability when computing the empirical Fisher matrix for ViTs, we perform all computations involving $H$ and its inverse in 64-bit precision. For the layer-wise reconstruction loss, we use 32-bit precision, as we do not observe similar instability.

### 4.3. Results on Vision Transformers

We present in Table 1 the results for the DeiT models using *RobOP*-CAP on different robust formulations. We observe that ($\|\Delta\|_F \leq \gamma$ or $\|\Delta_{:,i}\|_2 \leq \sqrt{\gamma}$) shows the strongest gains, especially for larger ViTs, improving ImageNet-1K accuracy by up to 10 points over the baseline. We attribute this behavior to two main factors:

- Multiple sources contribute to uncertainty in $H$ (empirical Fisher approximation, no Hessian recomputation

during pruning, and calibration-set noise). A fixed radius uncertainty set on $H$ can capture these effects simultaneously without targeting a specific error source.

- Gradient-proportional bounds on $\|\Delta_{:,i}\|_2$ may be reasonable early in pruning, but at high sparsity, the gradients can change substantially. Thus, $\|\Delta_{:,i}\|_2$ is likely not proportional to $\|\nabla \ell_i(\bar{w})\|_2$ anymore, making a constant bound on $\|\Delta_{:,i}\|_2$ better suited.

| Sparsity | Uncertainty Set | DeiT Tiny | DeiT Small | DeiT Base |
|---|---|---|---|---|
| | Baseline | $68.30 \pm 0.07$ | $76.70 \pm 0.10$ | $79.48 \pm 0.04$ |
| 0.5 | $\|\Delta\|_F \leq \gamma$ or $\|\Delta_{:,i}\|_2 \leq \sqrt{\gamma}$ | $68.32 \pm 0.05$ | $\mathbf{77.72 \pm 0.08}$ | $80.18 \pm 0.03$ |
| | $\|\Delta_{:,i}\|_2 \leq (\sqrt{\gamma}/N^{1/4}) \|\nabla \ell_i(\bar{w})\|_2$ or $\|\Delta\|_F \leq \gamma \mathrm{Tr}(H)/\sqrt{N}$ | $68.39 \pm 0.08$ | $77.34 \pm 0.09$ | $79.58 \pm 0.03$ |
| | $|\mu_i| \leq \gamma\sqrt{\frac{\eta_i \mathrm{Tr}(H)}{N}}$ | $\mathbf{68.41 \pm 0.08}$ | $77.54 \pm 0.09$ | $\mathbf{80.20 \pm 0.06}$ |
| | Baseline | $62.05 \pm 0.09$ | $71.36 \pm 0.23$ | $76.35 \pm 0.13$ |
| 0.6 | $\|\Delta\|_F \leq \gamma$ or $\|\Delta_{:,i}\|_2 \leq \sqrt{\gamma}$ | $62.12 \pm 0.08$ | $\mathbf{74.23 \pm 0.08}$ | $\mathbf{78.86 \pm 0.01}$ |
| | $\|\Delta_{:,i}\|_2 \leq (\sqrt{\gamma}/N^{1/4}) \|\nabla \ell_i(\bar{w})\|_2$ or $\|\Delta\|_F \leq \gamma \mathrm{Tr}(H)/\sqrt{N}$ | $\mathbf{62.20 \pm 0.12}$ | $73.17 \pm 0.15$ | $76.89 \pm 0.08$ |
| | $|\mu_i| \leq \gamma\sqrt{\frac{\eta_i \mathrm{Tr}(H)}{N}}$ | $62.15 \pm 0.10$ | $73.39 \pm 0.04$ | $77.30 \pm 0.06$ |
| | Baseline | $43.07 \pm 0.53$ | $53.77 \pm 0.88$ | $68.98 \pm 0.19$ |
| 0.7 | $\|\Delta\|_F \leq \gamma$ or $\|\Delta_{:,i}\|_2 \leq \sqrt{\gamma}$ | $43.99 \pm 0.19$ | $\mathbf{60.04 \pm 0.32}$ | $\mathbf{73.90 \pm 0.04}$ |
| | $\|\Delta_{:,i}\|_2 \leq (\sqrt{\gamma}/N^{1/4}) \|\nabla \ell_i(\bar{w})\|_2$ or $\|\Delta\|_F \leq \gamma \mathrm{Tr}(H)/\sqrt{N}$ | $44.10 \pm 0.24$ | $57.91 \pm 0.56$ | $70.08 \pm 0.18$ |
| | $|\mu_i| \leq \gamma\sqrt{\frac{\eta_i \mathrm{Tr}(H)}{N}}$ | $\mathbf{44.28 \pm 0.29}$ | $56.80 \pm 1.17$ | $70.65 \pm 0.24$ |
| | Baseline | $8.22 \pm 0.62$ | $12.74 \pm 1.12$ | $45.19 \pm 0.16$ |
| 0.8 | $\|\Delta\|_F \leq \gamma$ or $\|\Delta_{:,i}\|_2 \leq \sqrt{\gamma}$ | $8.34 \pm 0.79$ | $13.04 \pm 1.02$ | $\mathbf{55.19 \pm 0.18}$ |
| | $\|\Delta_{:,i}\|_2 \leq (\sqrt{\gamma}/N^{1/4}) \|\nabla \ell_i(\bar{w})\|_2$ or $\|\Delta\|_F \leq \gamma \mathrm{Tr}(H)/\sqrt{N}$ | $9.30 \pm 0.17$ | $\mathbf{14.73 \pm 0.85}$ | $47.36 \pm 0.66$ |
| | $|\mu_i| \leq \gamma\sqrt{\frac{\eta_i \mathrm{Tr}(H)}{N}}$ | $8.78 \pm 0.57$ | $13.87 \pm 0.10$ | $47.61 \pm 0.34$ |
| | Baseline | $51.93 \pm 0.35$ | $69.00 \pm 0.08$ | $75.29 \pm 0.06$ |
| 2:4 | $\|\Delta\|_F \leq \gamma$ or $\|\Delta_{:,i}\|_2 \leq \sqrt{\gamma}$ | $52.64 \pm 0.20$ | $70.48 \pm 0.01$ | $77.34 \pm 0.04$ |
| | $\|\Delta_{:,i}\|_2 \leq (\sqrt{\gamma}/N^{1/4}) \|\nabla \ell_i(\bar{w})\|_2$ or $\|\Delta\|_F \leq \gamma \mathrm{Tr}(H)/\sqrt{N}$ | $53.01 \pm 0.25$ | $70.28 \pm 0.05$ | $\mathbf{77.41 \pm 0.06}$ |
| | $|\mu_i| \leq \gamma\sqrt{\frac{\eta_i \mathrm{Tr}(H)}{N}}$ | $\mathbf{53.06 \pm 0.40}$ | $\mathbf{70.68 \pm 0.05}$ | $77.29 \pm 0.06$ |

*Table 1.* Test accuracy of DeiT models on ImageNet-1K after CAP/*RobOP*-CAP pruning under different robust formulations. Results are averaged over 3 seeds with standard errors.

### 4.4. Results on LLMs

We present in Table 2 the results for Llama 3.1-8B (Grattafiori et al., 2024) and Llama-2-13b-hf (Touvron et al., 2023b) using *RobOP*-ALPS (Meng et al., 2024a) on different robust formulations. Detailed results for each classification task are reported in Appendix B.1.

| Sparsity | Uncertainty Set | C4 ↓ | Wikitext2 ↓ | PTB ↓ | Mean Acc. ↑ |
|---|---|---|---|---|---|
| 0.6 | Baseline | $18.56 \pm 0.08$ | $\mathbf{13.94 \pm 0.15}$ | $18.74 \pm 0.14$ | $55.87 \pm 0.06$ |
|  | $\|\Delta\|_F \leq \gamma$ | $18.63 \pm 0.14$ | $13.97 \pm 0.27$ | $\mathbf{18.58 \pm 0.21}$ | $55.73 \pm 0.25$ |
|  | $\|\Delta\|_F \leq \gamma \frac{\mathrm{Tr}(XX^\top)}{\sqrt{N}}$ | $18.59 \pm 0.06$ | $14.03 \pm 0.10$ | $18.64 \pm 0.19$ | $55.93 \pm 0.19$ |
|  | $\|\mu_i\| \leq \gamma \sqrt{\frac{\xi_i \mathrm{Tr}(XX^\top)}{N}}$ | $\mathbf{18.55 \pm 0.13}$ | $14.00 \pm 0.23$ | $18.71 \pm 0.27$ | $\mathbf{56.07 \pm 0.26}$ |
| 0.7 | Baseline | $35.09 \pm 0.21$ | $30.16 \pm 0.66$ | $39.80 \pm 0.13$ | $48.86 \pm 0.22$ |
|  | $\|\Delta\|_F \leq \gamma$ | $34.55 \pm 0.59$ | $29.93 \pm 1.05$ | $40.06 \pm 0.60$ | $48.99 \pm 0.15$ |
|  | $\|\Delta\|_F \leq \gamma \frac{\mathrm{Tr}(XX^\top)}{\sqrt{N}}$ | $34.77 \pm 0.18$ | $29.64 \pm 0.52$ | $\mathbf{38.87 \pm 0.38}$ | $49.35 \pm 0.17$ |
|  | $\|\mu_i\| \leq \gamma \sqrt{\frac{\xi_i \mathrm{Tr}(XX^\top)}{N}}$ | $\mathbf{34.36 \pm 0.12}$ | $\mathbf{29.53 \pm 0.50}$ | $38.99 \pm 0.31$ | $\mathbf{49.39 \pm 0.07}$ |
| 0.8 | Baseline | $81.56 \pm 0.37$ | $113.18 \pm 6.66$ | $110.50 \pm 4.52$ | $\mathbf{41.14 \pm 0.29}$ |
|  | $\|\Delta\|_F \leq \gamma$ | $79.42 \pm 0.25$ | $105.01 \pm 6.25$ | $102.21 \pm 1.07$ | $40.75 \pm 0.04$ |
|  | $\|\Delta\|_F \leq \gamma \frac{\mathrm{Tr}(XX^\top)}{\sqrt{N}}$ | $\mathbf{78.82 \pm 0.72}$ | $\mathbf{103.44 \pm 4.19}$ | $101.91 \pm 2.89$ | $40.97 \pm 0.24$ |
|  | $\|\mu_i\| \leq \gamma \sqrt{\frac{\xi_i \mathrm{Tr}(XX^\top)}{N}}$ | $79.83 \pm 0.92$ | $107.23 \pm 4.26$ | $\mathbf{100.86 \pm 0.91}$ | $40.74 \pm 0.15$ |

*(a) Llama-3.1-8B*

| Sparsity | Uncertainty Set | C4 ↓ | Wikitext2 ↓ | PTB ↓ | Mean Acc. ↑ |
|---|---|---|---|---|---|
| 0.6 | Baseline | $9.82 \pm 0.04$ | $7.62 \pm 0.06$ | $104.41 \pm 2.49$ | $59.61 \pm 0.07$ |
|  | $\|\Delta\|_F \leq \gamma$ | $9.82 \pm 0.05$ | $7.67 \pm 0.06$ | $101.61 \pm 2.70$ | $\mathbf{59.85 \pm 0.11}$ |
|  | $\|\Delta\|_F \leq \gamma \frac{\mathrm{Tr}(XX^\top)}{\sqrt{N}}$ | $\mathbf{9.78 \pm 0.05}$ | $7.55 \pm 0.05$ | $\mathbf{100.38 \pm 3.07}$ | $59.51 \pm 0.31$ |
|  | $\|\mu_i\| \leq \gamma \sqrt{\frac{\xi_i \mathrm{Tr}(XX^\top)}{N}}$ | $9.79 \pm 0.05$ | $\mathbf{7.54 \pm 0.06}$ | $103.01 \pm 2.37$ | $\mathbf{59.85 \pm 0.19}$ |
| 0.7 | Baseline | $16.15 \pm 0.08$ | $14.53 \pm 0.24$ | $243.42 \pm 6.13$ | $54.26 \pm 0.06$ |
|  | $\|\Delta\|_F \leq \gamma$ | $15.94 \pm 0.08$ | $\mathbf{14.36 \pm 0.25}$ | $232.09 \pm 5.62$ | $54.02 \pm 0.11$ |
|  | $\|\Delta\|_F \leq \gamma \frac{\mathrm{Tr}(XX^\top)}{\sqrt{N}}$ | $15.99 \pm 0.11$ | $14.49 \pm 0.18$ | $235.45 \pm 5.39$ | $54.32 \pm 0.12$ |
|  | $\|\mu_i\| \leq \gamma \sqrt{\frac{\xi_i \mathrm{Tr}(XX^\top)}{N}}$ | $\mathbf{15.92 \pm 0.08}$ | $14.43 \pm 0.20$ | $\mathbf{232.30 \pm 6.51}$ | $\mathbf{54.37 \pm 0.31}$ |
| 0.8 | Baseline | $36.33 \pm 0.24$ | $41.92 \pm 2.10$ | $503.36 \pm 11.42$ | $42.80 \pm 0.31$ |
|  | $\|\Delta\|_F \leq \gamma$ | $\mathbf{35.14 \pm 0.31}$ | $39.80 \pm 1.81$ | $529.50 \pm 11.87$ | $42.95 \pm 0.45$ |
|  | $\|\Delta\|_F \leq \gamma \frac{\mathrm{Tr}(XX^\top)}{\sqrt{N}}$ | $35.26 \pm 0.29$ | $\mathbf{39.43 \pm 1.41}$ | $518.84 \pm 28.63$ | $42.68 \pm 0.19$ |
|  | $\|\mu_i\| \leq \gamma \sqrt{\frac{\xi_i \mathrm{Tr}(XX^\top)}{N}}$ | $35.22 \pm 0.42$ | $39.74 \pm 2.15$ | $514.66 \pm 23.08$ | $\mathbf{43.50 \pm 0.15}$ |

*(b) Llama-2-13b-hf*

*Table 2.* Test perplexity of Llama-3.1-8B (a) and Llama-2-13b-hf (b) on C4, WikiText2, and PTB, and mean accuracy on three classification tasks after ALPS/*RobOP*-ALPS pruning under different robust formulations. Results are averaged over 3 seeds with standard errors.

For LLMs, uncertainty sets that model the uncertainty induced by the finite number of samples seem to yield the strongest overall gains, improving the mean accuracy by up to 0.7 points and reducing WikiText2 downstream perplexity by up to 10 points.

The corresponding best values of $\gamma$ used in Tables 1 and 2 are reported in Appendix B.2.

### 4.5. Ablation on $N$

While Sections 4.3 and 4.4 focus on the standard sample sizes used in the literature (Meng et al., 2024a; Kuznedelev et al., 2023), we further investigate the impact of our framework in the low-$N$ regime, where only a limited number of samples is available. This setting is particularly challenging because estimation noise is larger, as suggested by Theorems 3.4 and 3.6. For LLMs, we reduce the calibration set to either 32 or 8 sequences of 2048 tokens. For ViTs, we additionally consider calibration sets of 2048, 1024, and 512 samples. The results are shown in Figure 1. As expected, we observe that the benefits of *RobOP* become increasingly pronounced as $N$ decreases and estimation noise grows, yielding gains of up to 33 WikiText2 perplexity points and 12 ImageNet-1K accuracy points. These results highlight the advantages of our framework in low-sample regimes.

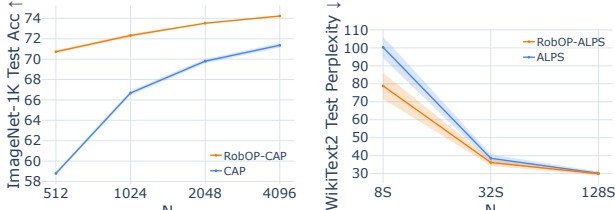

*(a)* DeiT-Small, 60% sparsity    *(b)* Llama-3.1-8B, 70% sparsity

*Figure 1.* Effect of the calibration set size $N$ on ImageNet-1K test accuracy for DeiT-Small pruned with CAP/*RobOP*-CAP at 60% sparsity (left), and on WikiText2 downstream perplexity for Llama-3.1-8B pruned with ALPS/*RobOP*-ALPS at 70% sparsity (right). For *RobOP*, we use the Frobenius-norm uncertainty set $\|\Delta\|_F \leq \gamma$.

In Figure 1, we report the performance obtained with *RobOP* using the Frobenius-norm uncertainty set $\|\Delta\|_F \leq \gamma$. The detailed results in Appendix B.3 further show that *RobOP* consistently improves performance across uncertainty sets and sparsity levels.

### 4.6. Efficiency of Robust Pruning

We note that all the uncertainty sets derived in the previous section result in adding a PSD matrix to $H$. In practice, computing this PSD term incurs negligible overhead relative to the baseline. For Frobenius-norm bounds on $\Delta$, or column-wise bounds on $\Delta_{:,i}$, the PSD regularizer is $\gamma I$ or $\gamma \frac{\mathrm{Tr}(H)}{\sqrt{N}} I$, which is computable at essentially no extra cost (at most a trace). The only nontrivial overhead arises with the eigenvalue-based uncertainty set, where the regularizer requires an SVD decomposition of $H$. For layer-wise reconstruction methods, $H$ is block-diagonal with the same block along the diagonal, $H = \mathrm{Diag}(XX^\top, \ldots, XX^\top)$, so this reduces to an SVD of a single block $XX^\top$. For Fisher-based methods, $H$ typically lacks this structure, making the eigenvalue-based regularization term more costly; empirically, however, the best results for larger ViTs use $\gamma I$, adding no overhead.

We report in Table 3 the pruning times for CAP on the DeiT models and ALPS on the Llama models under each uncertainty set, together with the original pruning time.

### 4.7. Importance of Robust Pruning

In this section, we show that the regularization terms motivated by *RobOP* and sometimes used in the literature without tuning (Singh & Alistarh, 2020a; Kuznedelev et al., 2023; Frantar & Alistarh, 2023; Meng et al., 2024a) do not merely improve numerical stability, but have a direct impact on generalization. Figures 2 and 3 highlight the importance of tuning the uncertainty set size parameter $\gamma$. Additional ablations are provided in Appendix B.4.

| Model | Uncertainty Set | Pruning time (minutes) |
|---|---|---|
| DeiT-Base | Baseline | 45.58 ± 2.22 |
| | $\|\Delta\|_F \leq \gamma$ or $\|\Delta_{:,i}\| \leq \sqrt{\gamma}$ | 45.71 ± 3.12 |
| | $\|\Delta_{:,i}\| \leq (\sqrt{\gamma}/N^{1/4})\|\nabla\ell_i(\bar{w})\|_2$ or $\|\Delta\|_F \leq \gamma \operatorname{Tr}(H)/\sqrt{N}$ | **44.15 ± 2.01** |
| | $|\mu_i| \leq \gamma\sqrt{\frac{\eta_i \operatorname{Tr}(H)}{N}}$ | 81.51 ± 1.64 |
| DeiT-Small | Baseline | **9.82 ± 0.38** |
| | $\|\Delta\|_F \leq \gamma$ or $\|\Delta_{:,i}\| \leq \sqrt{\gamma}$ | 10.02 ± 0.55 |
| | $\|\Delta_{:,i}\| \leq (\sqrt{\gamma}/N^{1/4})\|\nabla\ell_i(\bar{w})\|_2$ or $\|\Delta\|_F \leq \gamma \operatorname{Tr}(H)/\sqrt{N}$ | 10.11 ± 0.47 |
| | $|\mu_i| \leq \gamma\sqrt{\frac{\eta_i \operatorname{Tr}(H)}{N}}$ | 19.37 ± 0.62 |
| DeiT-Tiny | Baseline | 4.40 ± 0.02 |
| | $\|\Delta\|_F \leq \gamma$ or $\|\Delta_{:,i}\| \leq \sqrt{\gamma}$ | **4.37 ± 0.03** |
| | $\|\Delta_{:,i}\| \leq (\sqrt{\gamma}/N^{1/4})\|\nabla\ell_i(\bar{w})\|_2$ or $\|\Delta\|_F \leq \gamma \operatorname{Tr}(H)/\sqrt{N}$ | 4.38 ± 0.02 |
| | $|\mu_i| \leq \gamma\sqrt{\frac{\eta_i \operatorname{Tr}(H)}{N}}$ | 6.59 ± 0.02 |
| Llama-3.1-8B | Baseline | 53.97 ± 0.11 |
| | $\|\Delta\|_F \leq \gamma$ | 54.04 ± 0.09 |
| | $\|\Delta\|_F \leq \gamma \frac{\operatorname{Tr}(XX^\top)}{\sqrt{N}}$ | **53.90 ± 0.02** |
| | $|\mu_i| \leq \gamma\sqrt{\frac{\xi_i \operatorname{Tr}(XX^\top)}{N}}$ | 56.52 ± 0.08 |
| Llama-2-13b-hf | Baseline | 35.46 ± 0.11 |
| | $\|\Delta\|_F \leq \gamma$ | **35.38 ± 0.03** |
| | $\|\Delta\|_F \leq \gamma \frac{\operatorname{Tr}(XX^\top)}{\sqrt{N}}$ | 36.10 ± 0.58 |
| | $|\mu_i| \leq \gamma\sqrt{\frac{\xi_i \operatorname{Tr}(XX^\top)}{N}}$ | 40.10 ± 0.49 |

*Table 3.* Pruning time across uncertainty sets and architectures. We prune the DeiT models using CAP and the Llama models using ALPS. CAP can prune at multiple sparsity levels (e.g. $50\%, 60\%, 70\%$ and $80\%$) in a single run. For ALPS we report pruning times at $60\%$ sparsity. Results are averaged over three seeds with standard errors.

For CAP, the method implicitly includes a default regularization term $\gamma I$ with $\gamma = 10^{-8}$. Figure 2 shows that this default choice is suboptimal for DeiT-Base at $80\%$ sparsity.

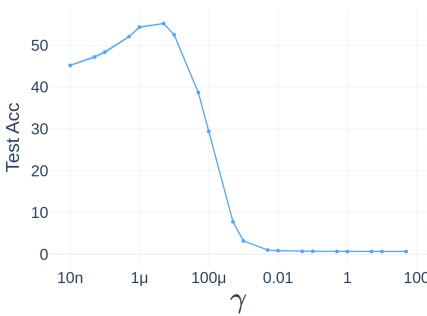

*Figure 2.* Effect of the uncertainty set size parameter $\gamma$ on the ImageNet-1K test accuracy of DeiT-Base, pruned to 80% sparsity using *RobOP*-CAP, under the uncertainty set $\|\Delta\|_F \leq \gamma$.

Similarly, ALPS adds $\gamma \frac{\operatorname{Tr}(XX^\top)}{p}$ to $H$ for numerical stability. While this differs from $\gamma \frac{\operatorname{Tr}(XX^\top)}{\sqrt{N}}$ (since $p$ changes across layers), it resembles this regularization term that arises with uncertainty in $H$ due to finite-sample estimation. Figure 3 shows a U-shaped trend in WikiText2 perplexity as a function of $\gamma$, indicating that neither very small nor very large values perform best. This highlights the practical

importance of tuning $\gamma$ (i.e., the size of the uncertainty set) for the layer-wise reconstruction loss as well.

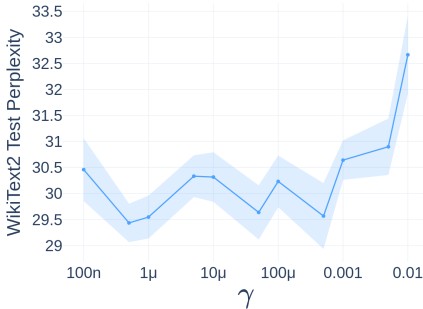

*Figure 3.* Effect of the uncertainty set size parameter $\gamma$ on the WikiText2 downstream perplexity of Llama-3.1-8B, pruned to 70% sparsity using *RobOP*-ALPS, under the uncertainty set $\|\Delta\|_F \leq \gamma \frac{\operatorname{Tr}(XX^\top)}{\sqrt{N}}$.

# 5. Conclusion

We introduced *RobOP*, a robust optimization framework for one-shot pruning. *RobOP* is flexible, supports multiple theory-driven uncertainty sets, and integrates efficiently into state-of-the-art pruning methods with minimal computational overhead. By explicitly modeling uncertainty in the pruning objective used by state-of-the-art baselines, *RobOP* consistently improves pruning performance. Beyond these empirical gains, *RobOP* provides a principled foundation for robust pruning under realistic sources of uncertainty. Finally, its ability to scale to ViTs and LLMs makes *RobOP* a practical and compelling choice for real-world applications.

# 6. Limitations and Future Work

While our study focuses on unstructured and semi-structured pruning, investigating structured pruning (Meng et al., 2024c) would be an interesting direction for future work. Other directions include sparse plus low-rank decompositions (Makni et al., 2025; 2026) of the weight matrix, transposable N:M sparsity (Meng et al., 2026), and weighted combinations of different proxy loss functions measuring network utility (Afriat et al., 2026), etc. Our results also show that performance is sensitive to the size of the uncertainty set, which we currently select via grid search. Developing principled methods for choosing $\gamma$ automatically would improve efficiency. Finally, our concentration bounds rely on independence assumptions, which may be restrictive for large language models due to token dependencies. Deriving bounds and uncertainty sets that relax this assumption would be a promising direction for future research.

## Acknowledgment

This work started during Gabriel Afriat's research internship at Google Research in summer 2024. Some subsequent research work was done at MIT. Rahul Mazumder acknowledges research funding from Office of Naval Research (ONR N000142512504).

## Impact Statement

This paper presents work whose goal is to advance the field of Machine Learning. There are many potential societal consequences of our work, none which we feel must be specifically highlighted here.

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

# A. Proofs

We consider the same assumption as Bunea & Xiao (2015):

**Assumption A.1.** ((Bunea & Xiao, 2015)) For a zero-mean sub-Gaussian random vector $X \in \mathbb{R}^p$, we assume that there exists a constant $c_0 > 0$ such that $E[(u^\top X)^2] \geq c_0 \|u^\top X\|_{\psi_2}^2$ for all $u \in \mathbb{R}^p$.

## A.1. Proof of Theorem 3.1

To show Theorem 3.1, we simplify the inner maximization problem in (5).

Let $p$ be the number of weights in the optimization problem ($\bar{w}, d \in \mathbb{R}^p$). Let's fix $d \in \mathbb{R}^p$ and consider $\Delta \in \mathbb{R}^{p \times p}$ with $\|\Delta\|_F \leq \gamma$. We have that:

$$
\begin{aligned}
d^\top \Delta d &= \langle d, \Delta d \rangle \\
&\leq \|d\|_2 . \|\Delta d\|_2 \quad \text{using Cauchy-Schwarz} \\
&\leq \|\Delta\|_{\text{op}} \|d\|_2^2 \\
&\leq \|\Delta\|_F \|d\|_2^2 \\
&\leq \gamma \|d\|_2^2 \quad \text{because} \|\Delta\|_F \leq \gamma
\end{aligned}
$$

Thus, we have that:

$$
\max_{\|\Delta\|_F \leq \gamma} \frac{1}{2} d^\top (H + \Delta) d \leq \gamma \|d\|_2^2
$$

Let's now consider $\Delta_0 = \frac{\gamma}{\|d\|_2^2} d d^\top$. We have that $\|\Delta_0\|_F = \frac{\gamma}{\|d\|_2^2} \|dd^\top\|_F = \frac{\gamma}{\|d\|_2^2} \|d\|_2^2 = \gamma$ and:

$$
\begin{aligned}
d^\top \Delta_0 d &= \frac{\gamma}{\|d\|_2^2} d^\top d d^\top d \\
&= \frac{\gamma}{\|d\|_2^2} \|d\|_2^4 \\
&= \gamma \|d\|_2^2
\end{aligned}
$$

In conclusion, we have that:

$$
\max_{\|\Delta\|_F \leq \gamma} \frac{1}{2} d^\top (H + \Delta) d = \gamma \|d\|_2^2
$$

And:

$$
\begin{array}{ll}
\min_{d} \max_{\|\Delta\|_F \leq \gamma} \frac{1}{2} d^\top (H + \Delta) d & = \quad \min_{d} \frac{1}{2} d^\top (H + \gamma I) d \\
s.t. \quad \bar{w} + d \in \mathcal{S} & \quad\quad s.t. \quad \bar{w} + d \in \mathcal{S}
\end{array}
$$

## A.2. Proof of Theorem 3.3

**Proof of Proposition 3.2**.

To show Proposition 3.2, we simplify the inner maximization problem in (7).

Let $p$ be the number of weights in the optimization problem ($\bar{w}, d \in \mathbb{R}^p$). Let's fix $d \in \mathbb{R}^p$ and consider $\Delta \in \mathbb{R}^{p \times N}$. We

denote as $\Delta_{:,i}$ and $A_{:,i}$ the $i$-th column of $\Delta$ and $A$ respectively, and we assume that $\|\Delta_{:,i}\|_2 \leq \epsilon_i$ for all $i \in [N]$.

$$
\begin{aligned}
\frac{1}{2N} d^\top (A+\Delta)(A+\Delta)^\top d &= \frac{1}{2N} \|(A+\Delta)^\top d\|_2^2 \\
&= \frac{1}{2N} \sum_{i=1}^N [(A+\Delta)^\top d]_i^2 \\
&= \frac{1}{2N} \sum_{i=1}^N \left( A_{:,i}^\top d + \Delta_{:,i}^\top d \right)^2 \\
&\leq \frac{1}{2N} \sum_{i=1}^N \left( |A_{:,i}^\top d| + \|\Delta_{:,i}\|_2 \|d\|_2 \right)^2 \quad \text{using Cauchy-Schwarz} \\
&\leq \frac{1}{2N} \sum_{i=1}^N \left( |A_{:,i}^\top d| + \epsilon_i \|d\|_2 \right)^2 = \frac{1}{2N} \| |A^\top d| + \|d\|_2 \epsilon \|_2^2
\end{aligned}
$$

Let's consider $\Delta_0$ such that $[\Delta_0]_{:,i} = \epsilon_i \text{sign}(A_{:,i}^\top d) \frac{d}{\|d\|_2}$. We have that $\|[\Delta_0]_{:,i}\|_2 = \epsilon_i$ and:

$$
\begin{aligned}
\frac{1}{2N} d^\top (A+\Delta_0)(A+\Delta_0)^\top d &= \frac{1}{2N} \sum_{i=1}^N \left( A_{:,i}^\top d + [\Delta_0]_{:,i}^\top d \right)^2 \\
&= \frac{1}{2N} \sum_{i=1}^N \left( A_{:,i}^\top d + \epsilon_i \text{sign}(A_{:,i}^\top d) \frac{1}{\|d\|_2} d^\top d \right)^2 \\
&= \frac{1}{2N} \sum_{i=1}^N \left( A_{:,i}^\top d + \text{sign}(A_{:,i}^\top d) \epsilon_i \|d\|_2 \right)^2 \\
&= \frac{1}{2N} \sum_{i=1}^N \left( |A_{:,i}^\top d| + \epsilon_i \|d\|_2 \right)^2 = \frac{1}{2N} \| |A^\top d| + \|d\|_2 \epsilon \|_2^2
\end{aligned}
$$

Thus, we do have that:

$$
\begin{array}{ll}
\min_d \max_{\|\Delta_{:,i}\| \leq \epsilon_i} \frac{1}{2N} d^\top (A+\Delta)(A+\Delta)^\top d &= \frac{1}{2N} \| |A^\top d| + \|d\|_2 \epsilon \|_2^2 \\
\quad s.t. \quad \bar{w} + d \in \mathcal{S} & \quad s.t. \quad \bar{w} + d \in \mathcal{S}
\end{array}
$$

**Proof of Theorem 3.3**

We simply need to bound the objective in (3.2):

$$
\| |A^\top d| + \|d\|_2 \epsilon \|_2 \leq \|A^\top d\| + \|d\|_2 \|\epsilon\|_2 \quad \text{using Cauchy Schwarz}
$$

Therefore:

$$
\begin{aligned}
\| |A^\top d| + \|d\|_2 \epsilon \|_2^2 &\leq 2\|A^\top d\|^2 + 2\|d\|_2^2 \|\epsilon\|_2^2 \\
&\leq 2d^\top AA^\top d + 2d^\top d \|\epsilon\|_2^2 \\
&\leq 2d^\top \left( AA^\top + \|\epsilon\|_2^2 I \right) d
\end{aligned}
$$

After dividing by $\frac{1}{2N}$, we obtain:

$$
\frac{1}{2N} \| |A^\top d| + \|d\|_2 \epsilon \|_2^2 \leq d^\top \left( H + \frac{1}{N} \|\epsilon\|_2^2 I \right) d
$$

## A.3. Proof of Theorem 3.4

First, let's show that $\mathbb{E}[\|\hat{M} - M\|_F^2] \leq \frac{1}{N}\mathbb{E}[\|Y\|_2^4]$

We have that:

$$
\begin{aligned}
\mathbb{E}[\|\hat{M} - M\|_F^2] &= \mathbb{E}\left[\left\|\frac{1}{N}\sum_{i=1}^{N}(X_i X_i^\top - M)\right\|_F^2\right] \\
&= \frac{1}{N^2}\sum_{i=1}^{N}\mathbb{E}\left[\|X_i X_i^\top - M\|_F^2\right] \text{ by independence of the } X_i \text{ and because } \mathbb{E}[X_i X_i^\top - M] = 0 \\
&= \frac{1}{N}\mathbb{E}[\|YY^\top - M\|_F^2] \\
&= \frac{1}{N}\mathbb{E}\left[\|YY^\top\|_F^2 + \|M\|_F^2 - 2\langle YY^\top, M\rangle\right] \\
&= \frac{1}{N}\left(\mathbb{E}\left[\|YY^\top\|_F^2\right] + \|M\|_F^2 - 2\mathbb{E}\left[\langle YY^\top, M\rangle\right]\right) \\
&= \frac{1}{N}\left(\mathbb{E}\left[\|Y\|_2^4\right] - \|M\|_F^2\right) \text{ because } \mathbb{E}\left[\langle YY^\top, M\rangle\right] = \|M\|_F^2 \\
&\leq \frac{1}{N}\mathbb{E}[\|Y\|_2^4]
\end{aligned}
$$

Second, let's bound $\mathbb{E}[\|Y\|_2^4]$:

Let's define $T = Y - \mu$

We have that:

$$
\|Y\|_2^4 = \|T + \mu\|_2^4 = \|\mu\|_2^4 + 4\|\mu\|_2^2\mu^\top T + 2\|\mu\|_2^2\|T\|_2^2 + 4\mu^\top T\|T\|_2^2 + 4(\mu^\top T)^2 + \|T\|_2^4
$$

Thus:

$$
\begin{aligned}
\mathbb{E}[\|T + \mu\|_2^4] &= \|\mu\|_2^4 + 2\|\mu\|_2^2\mathbb{E}[\|T\|_2^2] + 4\mathbb{E}[\mu^\top T\|T\|_2^2] + 4\mathbb{E}[(\mu^\top T)^2] + \mathbb{E}[\|T\|_2^4] \quad \text{because } \mathbb{E}[T] = 0 \\
&= \|\mu\|_2^4 + 2\|\mu\|_2^2\text{Tr}(\Sigma) + 4\mathbb{E}[\mu^\top T\|T\|_2^2] + 4\mu^\top\Sigma\mu + \mathbb{E}[\|T\|_2^4] \\
&\leq \|\mu\|_2^4 + 6\|\mu\|_2^2\text{Tr}(\Sigma) + 4\mathbb{E}[\mu^\top T\|T\|_2^2] + \mathbb{E}[\|T\|_2^4] \quad \text{because } \mu^\top\Sigma\mu \leq \|\Sigma\|_{\text{op}}\|\mu\|_2^2 \leq \text{Tr}(\Sigma)\|\mu\|_2^2
\end{aligned}
$$

Let's bound $\mathbb{E}[\|T\|_2^4]$:

$$
\begin{aligned}
\mathbb{E}[\|T\|_2^4] &= \mathbb{E}[(\|T\|_2^2)^2] \\
&= \mathbb{E}\left[\left(\sum_{i=1}^p T_i^2\right)\left(\sum_{j=1}^p T_i^2\right)\right] \\
&= \sum_{1 \leq i,j \leq p} \mathbb{E}[T_i^2 T_j^2] \\
&\leq \sum_{1 \leq i,j \leq p} \sqrt{\mathbb{E}[T_i^4]\mathbb{E}[T_j^4]} \quad \text{using Cauchy-Schwarz} \\
&\leq 16 \sum_{1 \leq i,j \leq p} \sqrt{\|T_i\|_{\psi_2}^4 \|T_j\|_{\psi_2}^4} \quad \text{by definition of } \|T_i\|_{\psi_2} = \sup_{k \geq 1} \frac{1}{k^{1/2}}(E[|T_i|^k])^{1/k} \\
&\leq \frac{16}{c_0^2} \sum_{1 \leq i,j \leq p} \mathbb{E}[T_i^2]\mathbb{E}[T_j^2] \quad \text{using Assumption A.1} \\
&\leq \frac{16}{c_0^2} \sum_{1 \leq i,j \leq p} \Sigma_{ii}\Sigma_{jj} \\
&\leq \frac{16}{c_0^2}\mathrm{Tr}(\Sigma)^2
\end{aligned}
$$

Let's bound $\mathbb{E}[\mu^\top T\|T\|_2^2]$:

$$
\begin{aligned}
\mathbb{E}[\mu^\top T\|T\|_2^2] &\leq \sqrt{\mathbb{E}[(\mu^\top T)^2\mathbb{E}[\|T\|_2^4]} \quad \text{using Cauchy-Schwarz} \\
&\leq \sqrt{\mu^\top \Sigma \mu}\frac{4}{c_0}\mathrm{Tr}(\Sigma) \quad \text{using the previously derived bound} \\
&\leq \frac{4}{c_0}\mathrm{Tr}(\Sigma)^{3/2}\|\mu\|_2 \\
&\leq \frac{4}{c_0}\mathrm{Tr}(\Sigma)\sqrt{\mathrm{Tr}(\Sigma)\|\mu\|_2^2} \\
&\leq \frac{4}{c_0}\mathrm{Tr}(\Sigma)\left(\mathrm{Tr}(\Sigma) + \|\mu\|_2^2\right) \quad \text{using } ab \leq (a+b)^2 \\
&\leq \frac{4}{c_0}\left(\mathrm{Tr}(\Sigma) + \|\mu\|_2^2\right)^2
\end{aligned}
$$

In summary, we have that:

$$
\mathbb{E}[\|Y\|_2^4] \leq \|\mu\|_2^4 + 6\|\mu\|_2^2\mathrm{Tr}(\Sigma) + \frac{16}{c_0}\left(\mathrm{Tr}(\Sigma) + \|\mu\|_2^2\right)^2 + \frac{16}{c_0}\mathrm{Tr}(\Sigma)^2
$$

Each term in the sum can be bounded by a factor of $\left(\mathrm{Tr}(\Sigma) + \|\mu\|_2^2\right)^2$. In addition, we know that $M = \Sigma + \mu\mu^\top$ so $\mathrm{Tr}(\Sigma) + \|\mu\|_2^2 = \mathrm{Tr}(M)$

Therefore, we have that:

$$
\mathbb{E}[\|Y\|_2^4] \lesssim \mathrm{Tr}(M)^2
$$

### A.4. Proof of Theorem 3.8

**Proof of Lemma 3.7.**

We have that:

$$
\begin{aligned}
\kappa_i &= \sqrt{\mathbb{E}[\|YY^\top u_i\|_2^2 - \lambda_i^2]} \\
&\leq \sqrt{\mathbb{E}[\|YY^\top u_i\|_2^2]} \\
&\leq \sqrt{\mathbb{E}[(Y^\top u_i)^2\|Y\|_2^2]} \quad \text{because } (Y^\top u_i) \in \mathbb{R} \\
&\leq \left(\mathbb{E}[(Y^\top u_i)^4]\mathbb{E}[\|Y\|_2^4]\right)^{\frac{1}{4}} \quad \text{(Cauchy-Schwarz)}
\end{aligned}
$$

First, let's bound $\mathbb{E}[(Y^\top u_i)^4]$:

Let $Z = Y^\top u_i$ and $W = Z - \mathbb{E}[Z] = u_i^\top(Y - \mu)$. Let $\mu = \mathbb{E}[Y]$.

$Z$ and $W$ are still sub-Gaussian since $Y$ is sub-Gaussian and:

$$
\begin{aligned}
\mathbb{E}[(Y^\top u_i)^4] &= \mathbb{E}[Z^4] \\
&= \mathbb{E}[(W + \mathbb{E}[Z])^4] \\
&= 16\mathbb{E}\left[\left(\frac{W + \mathbb{E}[Z]}{2}\right)^4\right] \\
&\leq 16\mathbb{E}\left[\frac{W^4 + \mathbb{E}[Z]^4}{2}\right] \quad \text{because } x \to x^4 \text{ is convex} \\
&\leq 8\mathbb{E}[W^4] + 8\mathbb{E}[Z]^4
\end{aligned}
$$

By definition, $\|W\|_{\psi_2} = \sup_{k\geq 1} \frac{1}{k^{1/2}}(E[|W|^k])^{1/k}$. Therefore, $16\|W\|_{\psi_2}^4 \geq E[W^4]$. As seen in (Bunea & Xiao, 2015), we have under Assumption A.1 that $u^\top\Sigma u \geq c_0\|W\|_\psi^2$ with $\Sigma$ the covariance matrix of $Y$. Thus, we find that:

$$
\begin{aligned}
\mathbb{E}[W^4] &\leq \frac{16}{c_0^2}(u_i^\top\Sigma u_i)^2 \\
&\leq \frac{16}{c_0^2}(u_i^\top M u_i)^2 \quad \text{because } M = \Sigma + \mu\mu^\top \succeq \Sigma \\
&\leq \frac{16}{c_0^2}\lambda_i^2
\end{aligned}
$$

We also have that

$$
\lambda_i = u_i^\top\Sigma u_i + u_i^\top\mu\mu^\top u_i \geq u_i^\top\mu\mu^\top u_i = (\mu^\top u_i)^2 \quad \text{because } \Sigma \succeq 0
$$

Thus:

$$
\mathbb{E}[Z]^4 = (\mu^\top u_i)^4 \leq \lambda_i^2
$$

This gives us that

$$
\mathbb{E}[(Y^\top u_i)^4] \lesssim \lambda_i^2
$$

Second, let's bound $\mathbb{E}[\|Y\|_2^4]$:

As seen in the proof of Theorem 3.4 above, we have that:

$$
\mathbb{E}[\|Y\|_2^4] \lesssim \mathrm{Tr}(M)^2
$$

Finally, we obtain:

$$\kappa_i \leq \left(\mathbb{E}[(Y^\top u_i)^4]\mathbb{E}[\|Y\|_2^4]\right)^{\frac{1}{4}}$$
$$\lesssim \left(\lambda_i^2 \mathrm{Tr}(M)^2\right)^{\frac{1}{4}} = \sqrt{\lambda_i \mathrm{Tr}(M)}$$

**Proof of Theorem 3.8.**

We need to bound the objective in (15).

Let $p$ be the number of weights in the optimization problem ($\bar{w}, d \in \mathbb{R}^p$). Let's fix $d \in \mathbb{R}^p$. Let's decompose $d$ in $d_{\mathrm{out}}$ blocks of size $d_{\mathrm{in}}$: $d = (d^{(1)}, \ldots, d^{(d_{\mathrm{out}})})^\top$.

We have that:

$$\max_{|\mu_i| \leq \gamma\sqrt{\frac{\xi_i \mathrm{Tr}(XX^\top)}{N}}} d^\top U \mathrm{Diag}(D(\mu), \ldots, D(\mu))U^\top d = \max_{|\mu_i| \leq \gamma\sqrt{\frac{\xi_i \mathrm{Tr}(XX^\top)}{N}}} \sum_{k=1}^{d_{\mathrm{out}}} d^{(k)^\top} P D(\mu) P^\top d^{(k)}$$

$$= \sum_{k=1}^{d_{\mathrm{out}}} \max_{|\mu_i| \leq \gamma\sqrt{\frac{\xi_i \mathrm{Tr}(XX^\top)}{N}}} d^{(k)^\top} P D(\mu) P^\top d^{(k)}$$

Let $k \in [d_{\mathrm{out}}]$. We note $u^{(k)} = P^\top d^{(k)}$.

$$d^{(k)^\top} P \mathrm{Diag}(\mu_1, \ldots, \mu_{d_{\mathrm{in}}}) P^\top d^{(k)} = u^{(k)^\top} \mathrm{Diag}(\mu_1, \ldots, \mu_{d_{\mathrm{in}}}) u^{(k)}$$

$$= \sum_{i=1}^{d_{\mathrm{in}}} \mu_i [u^{(k)}]_i^2$$

Therefore:

$$\max_{|\mu_i| \leq \gamma\sqrt{\frac{\xi_i \mathrm{Tr}(XX^\top)}{N}}} d^\top P \mathrm{Diag}(\mu_1, \ldots, \mu_{d_{\mathrm{in}}}) P^\top d = \max_{|\mu_i| \leq \gamma\sqrt{\frac{\xi_i \mathrm{Tr}(XX^\top)}{N}}} \sum_{i=1}^{d_{\mathrm{in}}} \mu_i [u^{(k)}]_i^2$$

$$= \sum_{i=1}^{d_{\mathrm{in}}} \max_{|\mu_i| \leq \gamma\sqrt{\frac{\xi_i \mathrm{Tr}(XX^\top)}{N}}} \mu_i [u^{(k)}]_i^2$$

$$= \sum_{i=1}^{d_{\mathrm{in}}} \gamma\sqrt{\frac{\xi_i \mathrm{Tr}(XX^\top)}{N}} [u^{(k)}]_i^2$$

$$= \gamma\sqrt{\frac{\mathrm{Tr}(XX^\top)}{N}} u^{(k)^\top} \mathrm{Diag}(\sqrt{\xi_1}, \ldots, \sqrt{\xi_{d_{\mathrm{in}}}}) u^{(k)}$$

$$= \gamma\sqrt{\frac{\mathrm{Tr}(XX^\top)}{N}} d^{(k)^\top} P D(\xi) P^\top d^{(k)}$$

And:

$$\max_{|\mu_i| \leq \gamma\sqrt{\frac{\xi_i \mathrm{Tr}(XX^\top)}{N}}} d^\top U \mathrm{Diag}(D(\mu), \ldots, D(\mu) U^\top d = \gamma\sqrt{\frac{\mathrm{Tr}(XX^\top)}{N}} \sum_{k=1}^{d_{\mathrm{out}}} d^{(k)^\top} P D(\xi) P^\top d^{(k)}$$

$$= \gamma\sqrt{\frac{\mathrm{Tr}(XX^\top)}{N}} U \mathrm{Diag}(D(\sqrt{\xi}), \ldots, D(\sqrt{\xi})) U^\top d$$

# B. Additional Experimental Results

## B.1. Detailed results per classification task

| Sparsity | Unc. set | C4 ↓ | Wikitext2 ↓ | PTB ↓ | OpenBookQA ↑ | Winogrande ↑ | PIQA ↑ | Mean Acc. ↑ |
|---|---|---|---|---|---|---|---|---|
| 0.6 | Baseline | $18.56 \pm 0.08$ | $\mathbf{13.94 \pm 0.15}$ | $18.74 \pm 0.14$ | $26.00 \pm 0.22$ | $68.25 \pm 0.28$ | $73.36 \pm 0.27$ | $55.87 \pm 0.06$ |
| | $\|\Delta\|_F \leq \gamma$ | $18.63 \pm 0.14$ | $13.97 \pm 0.27$ | $\mathbf{18.58 \pm 0.21}$ | $25.53 \pm 0.66$ | $68.30 \pm 0.31$ | $73.36 \pm 0.27$ | $55.73 \pm 0.25$ |
| | $\|\Delta\|_F \leq \gamma \frac{\text{Tr}(XX^\top)}{\sqrt{N}}$ | $18.59 \pm 0.06$ | $14.03 \pm 0.10$ | $18.64 \pm 0.19$ | $26.00 \pm 0.07$ | $68.35 \pm 0.52$ | $\mathbf{73.45 \pm 0.14}$ | $55.93 \pm 0.19$ |
| | $|\mu_i| \leq \gamma \sqrt{\frac{\xi_i \text{Tr}(XX^\top)}{N}}$ | $\mathbf{18.55 \pm 0.13}$ | $14.00 \pm 0.23$ | $18.71 \pm 0.27$ | $\mathbf{26.20 \pm 0.42}$ | $\mathbf{68.72 \pm 0.50}$ | $73.30 \pm 0.22$ | $\mathbf{56.07 \pm 0.26}$ |
| 0.7 | Baseline | $35.09 \pm 0.21$ | $30.16 \pm 0.66$ | $39.80 \pm 0.13$ | $19.27 \pm 0.22$ | $61.64 \pm 0.62$ | $65.67 \pm 0.12$ | $48.86 \pm 0.22$ |
| | $\|\Delta\|_F \leq \gamma$ | $34.55 \pm 0.59$ | $29.93 \pm 1.05$ | $40.06 \pm 0.60$ | $19.73 \pm 0.37$ | $61.30 \pm 0.25$ | $65.92 \pm 0.33$ | $48.99 \pm 0.15$ |
| | $\|\Delta\|_F \leq \gamma \frac{\text{Tr}(XX^\top)}{\sqrt{N}}$ | $34.77 \pm 0.18$ | $29.64 \pm 0.52$ | $\mathbf{38.87 \pm 0.38}$ | $19.93 \pm 0.30$ | $\mathbf{61.98 \pm 0.19}$ | $66.12 \pm 0.34$ | $49.35 \pm 0.17$ |
| | $|\mu_i| \leq \gamma \sqrt{\frac{\xi_i \text{Tr}(XX^\top)}{N}}$ | $\mathbf{34.36 \pm 0.12}$ | $\mathbf{29.53 \pm 0.50}$ | $38.99 \pm 0.31$ | $\mathbf{20.27 \pm 0.04}$ | $61.54 \pm 0.20$ | $\mathbf{66.36 \pm 0.38}$ | $\mathbf{49.39 \pm 0.07}$ |
| 0.8 | Baseline | $81.56 \pm 0.37$ | $113.18 \pm 6.66$ | $110.50 \pm 4.52$ | $13.07 \pm 0.07$ | $\mathbf{52.46 \pm 0.57}$ | $57.91 \pm 0.42$ | $\mathbf{41.14 \pm 0.29}$ |
| | $\|\Delta\|_F \leq \gamma$ | $79.42 \pm 0.25$ | $105.01 \pm 6.25$ | $102.21 \pm 1.07$ | $13.27 \pm 0.27$ | $50.93 \pm 0.28$ | $58.05 \pm 0.27$ | $40.75 \pm 0.04$ |
| | $\|\Delta\|_F \leq \gamma \frac{\text{Tr}(XX^\top)}{\sqrt{N}}$ | $\mathbf{78.82 \pm 0.72}$ | $\mathbf{103.44 \pm 4.19}$ | $101.91 \pm 2.89$ | $\mathbf{13.60 \pm 0.12}$ | $51.25 \pm 0.43$ | $\mathbf{58.07 \pm 0.37}$ | $40.97 \pm 0.24$ |
| | $|\mu_i| \leq \gamma \sqrt{\frac{\xi_i \text{Tr}(XX^\top)}{N}}$ | $79.83 \pm 0.92$ | $107.23 \pm 4.26$ | $\mathbf{100.86 \pm 0.91}$ | $13.20 \pm 0.40$ | $51.14 \pm 0.44$ | $57.89 \pm 0.08$ | $40.74 \pm 0.15$ |

*Table 4.* Test perplexity of Llama-3.1-8B on C4, WikiText2, and PTB, and test accuracy on three classification tasks (OpenBookQA, Winogrande, PIQA) after *RobOP*-ALPS pruning under different robust formulations. Mean accuracy is averaged across the three classification tasks. Results are reported over 3 seeds with standard errors.

| Sparsity | Unc. set | C4 ↓ | Wikitext2 ↓ | PTB ↓ | OpenBookQA ↑ | Winogrande ↑ | PIQA ↑ | Mean Acc. ↑ |
|---|---|---|---|---|---|---|---|---|
| 0.6 | Baseline | $9.82 \pm 0.04$ | $7.62 \pm 0.06$ | $104.41 \pm 2.49$ | $30.87 \pm 0.07$ | $71.59 \pm 0.09$ | $76.39 \pm 0.17$ | $59.61 \pm 0.07$ |
| | $\|\Delta\|_F \leq \gamma$ | $9.82 \pm 0.05$ | $7.67 \pm 0.06$ | $101.61 \pm 2.70$ | $\mathbf{31.53 \pm 0.53}$ | $71.74 \pm 0.12$ | $76.28 \pm 0.36$ | $\mathbf{59.85 \pm 0.11}$ |
| | $\|\Delta\|_F \leq \gamma \frac{\text{Tr}(XX^\top)}{\sqrt{N}}$ | $\mathbf{9.78 \pm 0.05}$ | $7.55 \pm 0.05$ | $\mathbf{100.38 \pm 3.07}$ | $30.87 \pm 0.74$ | $71.38 \pm 0.22$ | $76.30 \pm 0.05$ | $59.51 \pm 0.31$ |
| | $|\mu_i| \leq \gamma \sqrt{\frac{\xi_i \text{Tr}(XX^\top)}{N}}$ | $9.79 \pm 0.05$ | $\mathbf{7.54 \pm 0.06}$ | $103.01 \pm 2.37$ | $30.67 \pm 0.58$ | $\mathbf{72.22 \pm 0.18}$ | $\mathbf{76.66 \pm 0.20}$ | $\mathbf{59.85 \pm 0.19}$ |
| 0.7 | Baseline | $16.15 \pm 0.08$ | $14.53 \pm 0.24$ | $243.42 \pm 6.13$ | $25.87 \pm 0.33$ | $66.43 \pm 0.39$ | $70.48 \pm 0.34$ | $54.26 \pm 0.06$ |
| | $\|\Delta\|_F \leq \gamma$ | $15.94 \pm 0.08$ | $\mathbf{14.36 \pm 0.25}$ | $\mathbf{232.09 \pm 5.62}$ | $25.33 \pm 0.33$ | $66.48 \pm 0.26$ | $70.24 \pm 0.03$ | $54.02 \pm 0.11$ |
| | $\|\Delta\|_F \leq \gamma \frac{\text{Tr}(XX^\top)}{\sqrt{N}}$ | $15.99 \pm 0.11$ | $14.49 \pm 0.18$ | $235.45 \pm 5.39$ | $\mathbf{26.27 \pm 0.41}$ | $66.01 \pm 0.39$ | $\mathbf{70.67 \pm 0.41}$ | $54.32 \pm 0.12$ |
| | $|\mu_i| \leq \gamma \sqrt{\frac{\xi_i \text{Tr}(XX^\top)}{N}}$ | $\mathbf{15.92 \pm 0.08}$ | $14.43 \pm 0.20$ | $232.30 \pm 6.51$ | $26.00 \pm 0.53$ | $\mathbf{66.72 \pm 0.30}$ | $70.38 \pm 0.40$ | $\mathbf{54.37 \pm 0.31}$ |
| 0.8 | Baseline | $36.33 \pm 0.24$ | $41.92 \pm 2.10$ | $503.36 \pm 11.42$ | $13.87 \pm 0.48$ | $55.17 \pm 0.27$ | $59.36 \pm 0.36$ | $42.80 \pm 0.31$ |
| | $\|\Delta\|_F \leq \gamma$ | $\mathbf{35.14 \pm 0.31}$ | $39.80 \pm 1.81$ | $529.50 \pm 11.87$ | $\mathbf{14.20 \pm 0.87}$ | $54.96 \pm 0.18$ | $59.70 \pm 0.43$ | $42.95 \pm 0.45$ |
| | $\|\Delta\|_F \leq \gamma \frac{\text{Tr}(XX^\top)}{\sqrt{N}}$ | $35.26 \pm 0.29$ | $\mathbf{39.43 \pm 1.41}$ | $518.84 \pm 28.63$ | $14.13 \pm 0.07$ | $54.49 \pm 0.47$ | $59.41 \pm 0.19$ | $42.68 \pm 0.19$ |
| | $|\mu_i| \leq \gamma \sqrt{\frac{\xi_i \text{Tr}(XX^\top)}{N}}$ | $35.22 \pm 0.42$ | $39.74 \pm 2.15$ | $514.66 \pm 23.08$ | $13.80 \pm 0.20$ | $\mathbf{56.59 \pm 0.05}$ | $\mathbf{60.12 \pm 0.31}$ | $\mathbf{43.50 \pm 0.15}$ |

*Table 5.* Test perplexity of Llama-2-13b-hf on C4, WikiText2, and PTB, and test accuracy on three classification tasks (OpenBookQA, Winogrande, PIQA) after *RobOP*-ALPS pruning under different robust formulations. Mean accuracy is averaged across the three classification tasks. Mean results are reported over 3 seeds with standard errors.

## B.2. Selected values of $\gamma$

The corresponding best values of $\gamma$ used in Tables 1 and 2 are reported in Tables 6 and 7 respectively. For vision models trained with the Fisher loss, the selected value of $\gamma$ remains relatively stable across sparsity levels and architectures, and appears to be driven primarily by the choice of uncertainty set. For the layer-wise reconstruction loss, $\gamma$ varies more across sparsity levels and models, further highlighting the need for careful tuning.

## B.3. Ablation study on $N$

Figure 1 shows the impact of *RobOP* on CAP and ALPS in the low-$N$ regime under the uncertainty set $\|\Delta\|_F \leq \gamma$. We show in Tables 8 and 9 the detailed results obtained across calibration size $N$, sparsity levels, and uncertainty sets for both DeiT-Small and Llama-3.1-8B respectively.

| Sparsity | Uncertainty Set | DeiT Tiny | DeiT Small | DeiT Base |
|---|---|---|---|---|
| 0.5 | Baseline | 1.00e-08 | 1.00e-08 | 1.00e-08 |
| | $\|\Delta\|_F \le \gamma$ or $\|\Delta_{:,i}\| \le \sqrt{\gamma}$ | 1.00e-07 | 5.00e-06 | 1.00e-07 |
| | $\|\Delta_{:,i}\| \le (\sqrt{\gamma}/N^{1/4}) \|\nabla\ell_i(\bar{w})\|_2$ or $\|\Delta\|_F \le \gamma \operatorname{Tr}(H)/\sqrt{N}$ | 0.001 | 0.010 | 0.001 |
| | $|\mu_i| \le \gamma\sqrt{\frac{\xi_i \operatorname{Tr}(H)}{N}}$ | 0.500 | 5.000 | 5.000 |
| 0.6 | Baseline | 1.00e-08 | 1.00e-08 | 1.00e-08 |
| | $\|\Delta\|_F \le \gamma$ or $\|\Delta_{:,i}\| \le \sqrt{\gamma}$ | 1.00e-05 | 1.00e-05 | 1.00e-06 |
| | $\|\Delta_{:,i}\| \le (\sqrt{\gamma}/N^{1/4}) \|\nabla\ell_i(\bar{w})\|_2$ or $\|\Delta\|_F \le \gamma \operatorname{Tr}(H)/\sqrt{N}$ | 0.005 | 0.050 | 0.001 |
| | $|\mu_i| \le \gamma\sqrt{\frac{\xi_i \operatorname{Tr}(H)}{N}}$ | 1.000 | 5.000 | 1.000 |
| 0.7 | Baseline | 1.00e-08 | 1.00e-08 | 1.00e-08 |
| | $\|\Delta\|_F \le \gamma$ or $\|\Delta_{:,i}\| \le \sqrt{\gamma}$ | 5.00e-05 | 5.00e-05 | 1.00e-06 |
| | $\|\Delta_{:,i}\| \le (\sqrt{\gamma}/N^{1/4}) \|\nabla\ell_i(\bar{w})\|_2$ or $\|\Delta\|_F \le \gamma \operatorname{Tr}(H)/\sqrt{N}$ | 0.010 | 0.005 | 0.001 |
| | $|\mu_i| \le \gamma\sqrt{\frac{\xi_i \operatorname{Tr}(H)}{N}}$ | 0.500 | 1.000 | 1.000 |
| 0.8 | Baseline | 1.00e-08 | 1.00e-08 | 1.00e-08 |
| | $\|\Delta\|_F \le \gamma$ or $\|\Delta_{:,i}\| \le \sqrt{\gamma}$ | 5.00e-05 | 1.00e-07 | 5.00e-06 |
| | $\|\Delta_{:,i}\| \le (\sqrt{\gamma}/N^{1/4}) \|\nabla\ell_i(\bar{w})\|_2$ or $\|\Delta\|_F \le \gamma \operatorname{Tr}(H)/\sqrt{N}$ | 0.010 | 0.005 | 0.001 |
| | $|\mu_i| \le \gamma\sqrt{\frac{\xi_i \operatorname{Tr}(H)}{N}}$ | 1.000 | 0.500 | 1.000 |
| 2:4 | Baseline | 1.00e-08 | 1.00e-08 | 1.00e-08 |
| | $\|\Delta\|_F \le \gamma$ or $\|\Delta_{:,i}\| \le \sqrt{\gamma}$ | 1.00e-04 | 1.00e-05 | 5.00e-06 |
| | $\|\Delta_{:,i}\| \le (\sqrt{\gamma}/N^{1/4}) \|\nabla\ell_i(\bar{w})\|_2$ or $\|\Delta\|_F \le \gamma \operatorname{Tr}(H)/\sqrt{N}$ | 0.010 | 0.050 | 0.050 |
| | $|\mu_i| \le \gamma\sqrt{\frac{\xi_i \operatorname{Tr}(H)}{N}}$ | 5.000 | 10.000 | 10.000 |

*Table 6.* Best value of $\gamma$ for the DeiT-models with $N = 4096$.

| Sparsity | Unc. set | Llama-3.1-8B | Llama-2-13b-hf |
|---|---|---|---|
| 0.6 | Baseline | 0.0100 | 0.0100 |
| | $\|\Delta\|_F \le \gamma$ | 5.00e-05 | 0.1000 |
| | $\|\Delta\|_F \le \gamma \frac{\operatorname{Tr}(XX^\top)}{\sqrt{N}}$ | 5.00e-05 | 0.0050 |
| | $|\mu_i| \le \gamma\sqrt{\frac{\xi_i \operatorname{Tr}(H)}{N}}$ | 0.1000 | 5.0000 |
| 0.7 | Baseline | 0.0100 | 0.0100 |
| | $\|\Delta\|_F \le \gamma$ | 1.00e-06 | 1.00e-07 |
| | $\|\Delta\|_F \le \gamma \frac{\operatorname{Tr}(XX^\top)}{\sqrt{N}}$ | 5.00e-05 | 5.00e-07 |
| | $|\mu_i| \le \gamma\sqrt{\frac{\xi_i \operatorname{Tr}(H)}{N}}$ | 1.00e-06 | 1.00e-07 |
| 0.8 | Baseline | 0.0100 | 0.0100 |
| | $\|\Delta\|_F \le \gamma$ | 1.0000 | 0.0001 |
| | $\|\Delta\|_F \le \gamma \frac{\operatorname{Tr}(XX^\top)}{\sqrt{N}}$ | 1.00e-06 | 5.00e-07 |
| | $|\mu_i| \le \gamma\sqrt{\frac{\xi_i \operatorname{Tr}(H)}{N}}$ | 5.00e-06 | 0.0001 |

*Table 7.* Best $\gamma$ values for the Llama models with $N$ corresponding to 128 sequences of 2048 tokens

| Sparsity | Method | 512 | 1024 | 2048 | 4096 |
|---|---|---|---|---|---|
| | Baseline | $72.48 \pm 0.07$ | $74.89 \pm 0.09$ | $76.03 \pm 0.06$ | $76.70 \pm 0.10$ |
| 0.5 | $\|\Delta\|_F \leq \gamma$ or $\|\Delta_{:,i}\| \leq \sqrt{\gamma}$ | $\mathbf{76.50 \pm 0.07}$ | $\mathbf{76.99 \pm 0.11}$ | $\mathbf{77.40 \pm 0.01}$ | $\mathbf{77.72 \pm 0.08}$ |
| | $\|\Delta_{:,i}\| \leq (\sqrt{\gamma}/N^{1/4}) \|\nabla \ell_i(\bar{w})\|_2$ or $\|\Delta\|_F \leq \gamma \operatorname{Tr}(H)/\sqrt{N}$ | $75.49 \pm 0.16$ | $76.55 \pm 0.08$ | $77.04 \pm 0.00$ | $77.34 \pm 0.09$ |
| | $|\mu_i| \leq \gamma \sqrt{\frac{\xi_i \operatorname{Tr}(H)}{N}}$ | $75.47 \pm 0.09$ | $76.54 \pm 0.05$ | $77.19 \pm 0.10$ | $77.54 \pm 0.09$ |
| | Baseline | $58.79 \pm 0.17$ | $66.68 \pm 0.22$ | $69.80 \pm 0.21$ | $71.36 \pm 0.23$ |
| 0.6 | $\|\Delta\|_F \leq \gamma$ or $\|\Delta_{:,i}\| \leq \sqrt{\gamma}$ | $\mathbf{70.73 \pm 0.13}$ | $\mathbf{72.32 \pm 0.15}$ | $\mathbf{73.52 \pm 0.03}$ | $\mathbf{74.23 \pm 0.08}$ |
| | $\|\Delta_{:,i}\| \leq (\sqrt{\gamma}/N^{1/4}) \|\nabla \ell_i(\bar{w})\|_2$ or $\|\Delta\|_F \leq \gamma \operatorname{Tr}(H)/\sqrt{N}$ | $64.82 \pm 0.38$ | $69.58 \pm 0.22$ | $71.60 \pm 0.18$ | $73.17 \pm 0.15$ |
| | $|\mu_i| \leq \gamma \sqrt{\frac{\xi_i \operatorname{Tr}(H)}{N}}$ | $67.45 \pm 0.32$ | $69.99 \pm 0.13$ | $72.17 \pm 0.10$ | $73.39 \pm 0.04$ |
| | Baseline | $18.48 \pm 1.71$ | $39.13 \pm 0.64$ | $47.20 \pm 0.17$ | $53.77 \pm 0.88$ |
| 0.7 | $\|\Delta\|_F \leq \gamma$ or $\|\Delta_{:,i}\| \leq \sqrt{\gamma}$ | $\mathbf{50.44 \pm 0.75}$ | $\mathbf{55.29 \pm 0.12}$ | $\mathbf{59.07 \pm 0.19}$ | $\mathbf{60.04 \pm 0.32}$ |
| | $\|\Delta_{:,i}\| \leq (\sqrt{\gamma}/N^{1/4}) \|\nabla \ell_i(\bar{w})\|_2$ or $\|\Delta\|_F \leq \gamma \operatorname{Tr}(H)/\sqrt{N}$ | $33.90 \pm 0.52$ | $45.87 \pm 0.53$ | $54.21 \pm 0.25$ | $57.91 \pm 0.56$ |
| | $|\mu_i| \leq \gamma \sqrt{\frac{\xi_i \operatorname{Tr}(H)}{N}}$ | $36.59 \pm 1.86$ | $46.52 \pm 1.32$ | $53.07 \pm 0.52$ | $56.80 \pm 1.17$ |
| | Baseline | $0.67 \pm 0.10$ | $4.00 \pm 0.37$ | $8.70 \pm 0.32$ | $12.74 \pm 1.12$ |
| 0.8 | $\|\Delta\|_F \leq \gamma$ or $\|\Delta_{:,i}\| \leq \sqrt{\gamma}$ | $\mathbf{6.20 \pm 0.47}$ | $\mathbf{9.95 \pm 0.66}$ | $10.39 \pm 0.88$ | $13.04 \pm 1.02$ |
| | $\|\Delta_{:,i}\| \leq (\sqrt{\gamma}/N^{1/4}) \|\nabla \ell_i(\bar{w})\|_2$ or $\|\Delta\|_F \leq \gamma \operatorname{Tr}(H)/\sqrt{N}$ | $2.99 \pm 0.10$ | $7.43 \pm 1.20$ | $\mathbf{12.91 \pm 0.15}$ | $\mathbf{14.73 \pm 0.85}$ |
| | $|\mu_i| \leq \gamma \sqrt{\frac{\xi_i \operatorname{Tr}(H)}{N}}$ | $4.58 \pm 0.73$ | $7.21 \pm 0.32$ | $11.99 \pm 0.97$ | $13.87 \pm 0.10$ |

*Table 8.* Test accuracy of DeiT-Small on ImageNet-1K after CAP and *RobOP*-CAP pruning under different robust formulations and $N$ regimes. Results are averaged over 3 seeds with standard errors.

| Dataset / Metric | Sparsity | Method | 8 $S$ | 32 $S$ | 128 $S$ |
|---|---|---|---|---|---|
| C4 Perplexity ↓ | 0.6 | Baseline | $27.58 \pm 1.04$ | $20.34 \pm 0.15$ | $18.56 \pm 0.12$ |
| | | $\|\Delta\|_F \leq \gamma$ | $24.41 \pm 0.52$ | $20.21 \pm 0.03$ | $18.63 \pm 0.14$ |
| | | $\|\Delta\|_F \leq \gamma \frac{\text{Tr}(XX^\top)}{\sqrt{N}}$ | $\mathbf{22.64 \pm 0.26}$ | $20.14 \pm 0.11$ | $18.59 \pm 0.10$ |
| | | $\|\mu_i\| \leq \gamma \sqrt{\frac{\xi_i \text{Tr}(XX^\top)}{N}}$ | $25.09 \pm 0.52$ | $\mathbf{19.87 \pm 0.11}$ | $\mathbf{18.55 \pm 0.13}$ |
| | 0.7 | Baseline | $77.47 \pm 6.15$ | $39.50 \pm 1.04$ | $35.09 \pm 0.34$ |
| | | $\|\Delta\|_F \leq \gamma$ | $62.02 \pm 3.46$ | $39.64 \pm 0.92$ | $34.55 \pm 0.59$ |
| | | $\|\Delta\|_F \leq \gamma \frac{\text{Tr}(XX^\top)}{\sqrt{N}}$ | $\mathbf{60.47 \pm 2.83}$ | $39.63 \pm 0.99$ | $34.77 \pm 0.29$ |
| | | $\|\mu_i\| \leq \gamma \sqrt{\frac{\xi_i \text{Tr}(XX^\top)}{N}}$ | $65.83 \pm 4.45$ | $\mathbf{39.45 \pm 0.73}$ | $\mathbf{34.36 \pm 0.19}$ |
| | 0.8 | Baseline | $229.94 \pm 19.90$ | $93.31 \pm 2.12$ | $81.56 \pm 0.37$ |
| | | $\|\Delta\|_F \leq \gamma$ | $\mathbf{156.68 \pm 11.58}$ | $90.35 \pm 3.17$ | $79.42 \pm 0.25$ |
| | | $\|\Delta\|_F \leq \gamma \frac{\text{Tr}(XX^\top)}{\sqrt{N}}$ | $167.99 \pm 8.49$ | $\mathbf{90.07 \pm 2.73}$ | $\mathbf{78.82 \pm 0.72}$ |
| | | $\|\mu_i\| \leq \gamma \sqrt{\frac{\xi_i \text{Tr}(XX^\top)}{N}}$ | $163.89 \pm 8.87$ | $90.65 \pm 1.95$ | $79.83 \pm 0.92$ |
| WikiText2 Perplexity ↓ | 0.6 | Baseline | $22.68 \pm 0.60$ | $15.86 \pm 0.54$ | $13.94 \pm 0.23$ |
| | | $\|\Delta\|_F \leq \gamma$ | $19.37 \pm 0.53$ | $15.46 \pm 0.43$ | $\mathbf{13.97 \pm 0.27}$ |
| | | $\|\Delta\|_F \leq \gamma \frac{\text{Tr}(XX^\top)}{\sqrt{N}}$ | $\mathbf{17.47 \pm 0.26}$ | $15.55 \pm 0.54$ | $14.03 \pm 0.15$ |
| | | $\|\mu_i\| \leq \gamma \sqrt{\frac{\xi_i \text{Tr}(XX^\top)}{N}}$ | $19.85 \pm 0.42$ | $\mathbf{15.10 \pm 0.45}$ | $14.00 \pm 0.23$ |
| | 0.7 | Baseline | $100.28 \pm 5.97$ | $38.51 \pm 2.22$ | $30.16 \pm 1.05$ |
| | | $\|\Delta\|_F \leq \gamma$ | $78.82 \pm 7.27$ | $\mathbf{36.15 \pm 1.80}$ | $29.93 \pm 1.05$ |
| | | $\|\Delta\|_F \leq \gamma \frac{\text{Tr}(XX^\top)}{\sqrt{N}}$ | $\mathbf{66.80 \pm 2.54}$ | $38.65 \pm 2.47$ | $29.64 \pm 0.82$ |
| | | $\|\mu_i\| \leq \gamma \sqrt{\frac{\xi_i \text{Tr}(XX^\top)}{N}}$ | $75.78 \pm 5.53$ | $38.82 \pm 2.88$ | $\mathbf{29.53 \pm 0.79}$ |
| | 0.8 | Baseline | $674.49 \pm 116.45$ | $171.44 \pm 19.79$ | $113.18 \pm 6.66$ |
| | | $\|\Delta\|_F \leq \gamma$ | $\mathbf{294.73 \pm 44.92}$ | $\mathbf{143.65 \pm 14.66}$ | $105.01 \pm 6.25$ |
| | | $\|\Delta\|_F \leq \gamma \frac{\text{Tr}(XX^\top)}{\sqrt{N}}$ | $346.10 \pm 55.59$ | $166.08 \pm 10.54$ | $\mathbf{103.44 \pm 4.19}$ |
| | | $\|\mu_i\| \leq \gamma \sqrt{\frac{\xi_i \text{Tr}(XX^\top)}{N}}$ | $345.72 \pm 30.45$ | $156.03 \pm 8.62$ | $107.23 \pm 4.26$ |
| PTB ↓ | 0.6 | Baseline | $28.50 \pm 0.64$ | $20.61 \pm 0.73$ | $18.74 \pm 0.23$ |
| | | $\|\Delta\|_F \leq \gamma$ | $25.00 \pm 0.58$ | $20.72 \pm 0.73$ | $\mathbf{18.58 \pm 0.21}$ |
| | | $\|\Delta\|_F \leq \gamma \frac{\text{Tr}(XX^\top)}{\sqrt{N}}$ | $\mathbf{23.82 \pm 0.56}$ | $\mathbf{20.39 \pm 0.59}$ | $18.64 \pm 0.30$ |
| | | $\|\mu_i\| \leq \gamma \sqrt{\frac{\xi_i \text{Tr}(XX^\top)}{N}}$ | $26.41 \pm 0.69$ | $20.55 \pm 0.65$ | $18.71 \pm 0.27$ |
| | 0.7 | Baseline | $144.12 \pm 6.94$ | $52.03 \pm 1.94$ | $39.80 \pm 0.21$ |
| | | $\|\Delta\|_F \leq \gamma$ | $\mathbf{89.19 \pm 2.14}$ | $51.13 \pm 1.33$ | $40.06 \pm 0.60$ |
| | | $\|\Delta\|_F \leq \gamma \frac{\text{Tr}(XX^\top)}{\sqrt{N}}$ | $91.86 \pm 3.44$ | $53.99 \pm 4.20$ | $\mathbf{38.87 \pm 0.61}$ |
| | | $\|\mu_i\| \leq \gamma \sqrt{\frac{\xi_i \text{Tr}(XX^\top)}{N}}$ | $110.87 \pm 5.00$ | $\mathbf{50.88 \pm 3.68}$ | $38.99 \pm 0.49$ |
| | 0.8 | Baseline | $701.06 \pm 60.99$ | $159.27 \pm 9.51$ | $110.50 \pm 4.52$ |
| | | $\|\Delta\|_F \leq \gamma$ | $\mathbf{352.17 \pm 45.14}$ | $\mathbf{149.25 \pm 6.66}$ | $102.21 \pm 1.07$ |
| | | $\|\Delta\|_F \leq \gamma \frac{\text{Tr}(XX^\top)}{\sqrt{N}}$ | $370.92 \pm 35.35$ | $161.46 \pm 6.02$ | $101.91 \pm 2.89$ |
| | | $\|\mu_i\| \leq \gamma \sqrt{\frac{\xi_i \text{Tr}(XX^\top)}{N}}$ | $419.89 \pm 18.73$ | $161.35 \pm 6.32$ | $\mathbf{100.86 \pm 0.91}$ |

*Table 9.* Test perplexity of Llama-3.1-8B on C4, WikiText2 and PTB after ALPS and *RobOP*-ALPS pruning under different robust formulations and $N$ regimes. Results are averaged over 3 seeds with standard errors. Here, $S$ denotes the sequence length, set to 2048.

## B.4. Ablation study on $\gamma$

For the Llama models with the regularization term $\gamma \frac{\text{Tr}(XX^\top)}{\sqrt{N}}$, we restrict the x-axis in the plots to $\gamma \in \left\{10^{-8}, 5 \times 10^{-8}, 10^{-7}, \dots, 10^{-1}\right\}$ to zoom in on the range of interest and more clearly illustrate the U-shaped trend, since perplexity increases rapidly beyond this point.

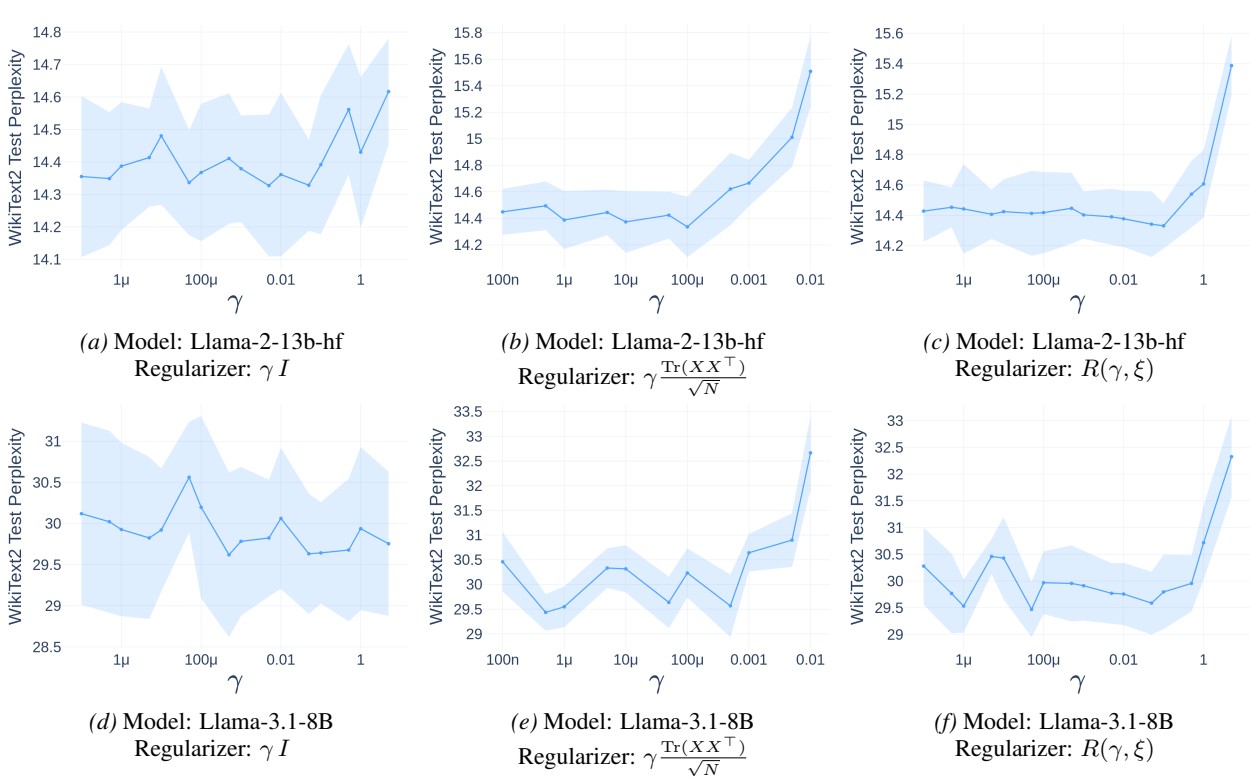

*Figure 4.* Effect of uncertainty size parameter $\gamma$ on downstream perplexity. We report WikiText2 perplexity at 70% sparsity for Llama-2-13b-hf (top row) and Llama-3.1-8B (bottom row) using *RobOP*-ALPS with different uncertainty sets. Mean perplexity is computed over 3 seeds with standard errors.

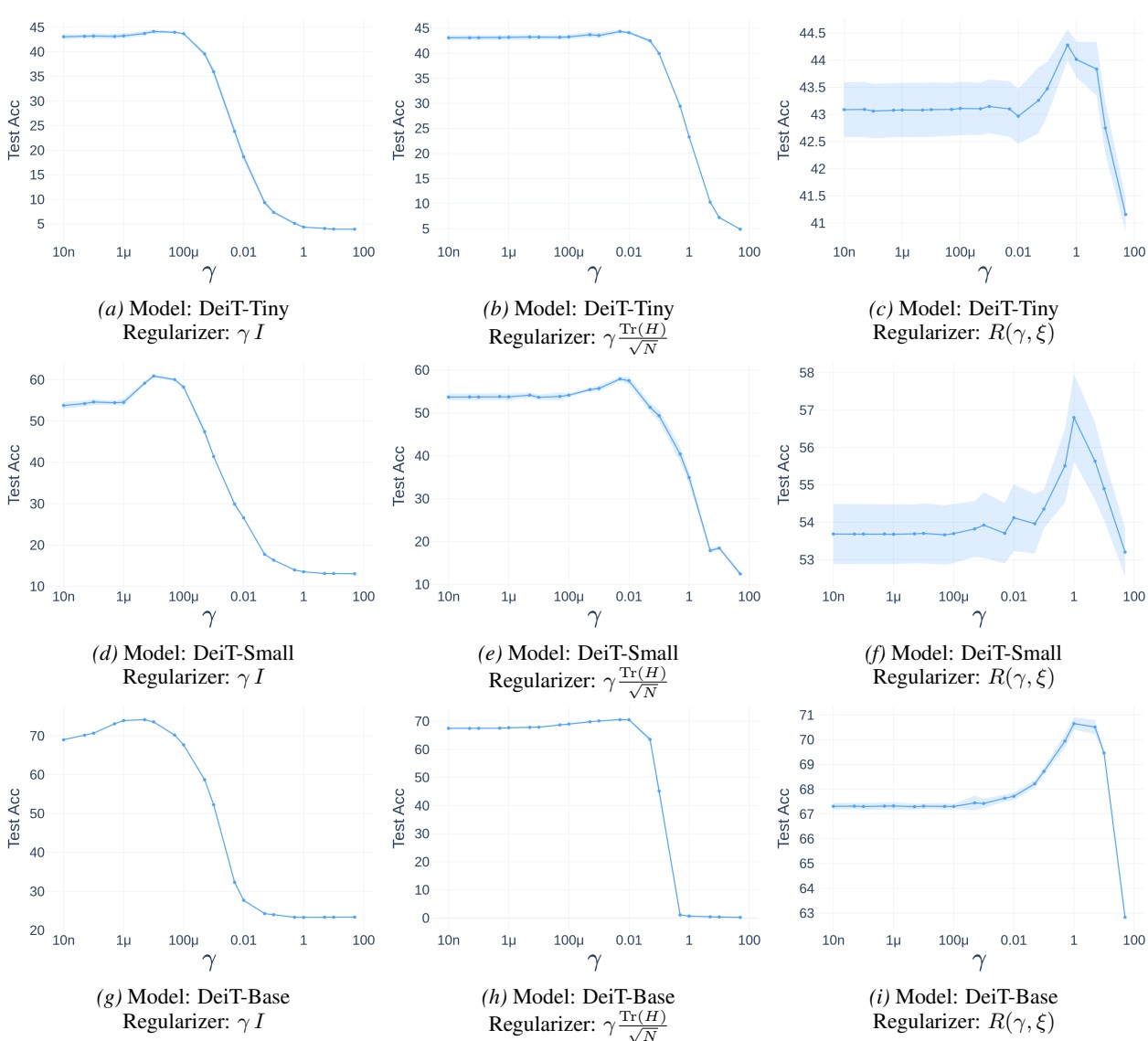

*Figure 5.* Effect of uncertainty size parameter $\gamma$ on ImageNet-1K test accuracy. We prune DeiT-Tiny (top row), DeiT-Small (middle row) and DeiT-Base (bottom row) at 70% sparsity using *RobOP*-CAP with different uncertainty sets. Mean accuracy is computed over 3 seeds with standard errors.

