# OpenReview forum: "A Robust Optimization Guided Pruning Framework for Vision and Large Language Models"
_ICML.cc/2026/Conference — ICML 2026 regular_

### Official Review · Reviewer_xXAq · 2026-03-09

**Soundness:** 2
**Presentation:** 3
**Significance:** 3
**Originality:** 2
**Overall Recommendation:** 4
**Confidence:** 4

**Summary:**

This paper introduces RobOP (Robust Optimization for One-shot Pruning), a novel framework designed to mitigate uncertainties in Hessian estimation that arise during one-shot pruning of large vision and language models. The authors identify key sources of uncertainty—such as finite calibration sets, the Fisher approximation, and outdated gradient estimates due to sequential pruning—and reformulate the standard quadratic pruning objective using robust optimization principles. By modeling uncertainty through different sets (e.g., bounded Frobenius norm or eigenvalue-wise perturbations), they derive equivalent convex objectives that introduce simple regularization terms. These robust variants integrate seamlessly into existing pruning methods like CAP for Vision Transformers and ALPS for LLMs. Experiments on DeiT models (ImageNet-1K) and Llama models (C4, WikiText2, PTB, and commonsense reasoning tasks) show consistent improvements, with accuracy gains up to 10 points and perplexity reductions of up to 10 points, all with minimal computational overhead.

**Compliance With Llm Reviewing Policy:**

Affirmed.

**Final Justification:**

I raised my score to 4.

**Key Questions For Authors:**

1、Regarding Theorem 3.3: You provide an upper bound rather than an exact equivalence for the per-sample gradient uncertainty set. How tight is this bound in practice? Could using the exact formulation (8) (solved via SOCP) ever yield better pruning performance than the approximate formulation (9), and if so, under what conditions? A response clarifying the practical gap between these formulations would help assess whether future work should pursue exact solutions.
2、Verification of Assumption A.1: Theorem 3.4 relies upon the assumption that the input follows a sub-Gaussian distribution. Has the author verified this assumption on the feature layer outputs of actual models? Should this assumption prove invalid, how might it affect the selection of the regularisation parameter γ?

**Limitations:**

yes

**Strengths And Weaknesses:**

Strengths and Weaknesses
Soundness: Technically rigorous with solid theoretical derivations and thorough experiments across multiple models, sparsity levels, and seeds. However, key assumptions (sub-Gaussian, eigenvector closeness) are unverified, and statistical significance is not established for observed gains.
Presentation: Exceptionally well-written with clear motivation and logical flow, properly situating the work in literature.
Significance: Addresses an important practical problem with substantial gains (up to 10 points) and immediate applicability via simple quadratic regularization. However, evaluation is narrow (only two methods, two families) and gains occur at sparsity levels that may be too high for practical deployment.
Originality: Offers a novel perspective applying robust optimization to Hessian uncertainty, with new concentration bounds. But the resulting regularizers closely resemble existing heuristics, offering explanation more than invention.

---

> ### Author Rebuttal · Authors · 2026-03-31
>
> We thank the reviewer for the careful reading and helpful comments.
>
> ## Assumptions of RobOP and statistical significance
>
> RobOP relies on standard covariance concentration assumptions [1,2,3]. We agree that their relevance to practical ViT/LLM settings should be discussed more clearly, and we will add this discussion to the paper.
>
> The sub-Gaussian assumption is reasonable here: ViT inputs are bounded by pixel ranges, and LLM inputs by the embedding weights, so layer inputs remain bounded. Since bounded random variables are sub-Gaussian, this assumption is well motivated.
>
> The independence assumption is more restrictive, since tokens in a sequence are not independent. We agree this is a limitation and will state it explicitly; extending the analysis to dependent data is an interesting direction for future work.
>
> We also agree that our eigenvalue uncertainty set assumes the eigenvectors of $M$ and $\hat M$ are close. We therefore compared the top-$k$ eigenvectors $C^{(k, 128)}$, $C^{(k, 512)}$ of ${X^{(128)}}^\top X^{(128)}$ and ${X^{(512)}}^\top X^{(512)}$, computed from 128 and 512 sequences of length 2048 respectively. Because the order of individual eigenvectors may be unstable, we compare subspaces via principal-angle cosines from the singular values of $D={C^{(k,128)}}^\top C^{(k,512)}$. We report below the mean principal-angle cosines for different values of $k$.
>
> |k|16|64|512|
> |-|-:|-:|-:|
> |0.mlp.down_proj|0.94|0.94|0.87|
> |0.mlp.gate_proj|0.99|0.97|0.93|
> |0.mlp.up_proj|0.99|0.97|0.93|
> |0.self_attn.k_proj|1.00|0.99|0.99|
> |0.self_attn.o_proj|0.97|0.96|0.95|
>
> The cosines are consistently close to 1, indicating strong subspace alignment and convergence (we report only the first layers because of space constraints). We agree this may weaken for much smaller $N$, and modeling eigenvector uncertainty is an important extension.
>
> Gains are statistically significant in most ViT settings and in many cases for LLMs. Additional experiments suggested by Reviewer 4AHW also show that RobOP becomes more advantageous as $H$ becomes noisier (when fewer samples are available): up to 32 accuracy points for RobOP-CAP and 33 perplexity points on WikiText2 for RobOP-ALPS. These gains are statistically significant.
>
> [1] Koltchinskii et al. (2014). Concentration inequalities and moment bounds for sample covariance operators.
>
> [2] Bunea et al. (2015). On the sample covariance matrix estimator of reduced effective rank population matrices, with applications to FPCA. Bernoulli, 21.
>
> [3] Loukas (2017). How close are the eigenvectors of the sample and actual covariance matrices? ICML.
>
> ## Evaluation and Gains
>
> Our goal was to show that RobOP improves strong one-shot pruning baselines. We considered two common pruning objectives, evaluated RobOP on 3 ViTs and 2 LLMs, and for LLMs report both perplexity and downstream classification.
>
> While further experiments would be valuable, we believe the current evaluation already shows that robust pruning is preferable to nominal pruning and that RobOP yields consistent practical gains.
>
> ## RobOP goes beyond a robust view of existing regularizers
>
> We agree that our robust formulations recover 2 existing regularizers. However, to the best of our knowledge, the other regularization terms we introduce are new. More importantly, RobOP does more than reinterpret existing regularizers. It provides a new robust pruning framework, introduces several uncertainty sets and robust formulations to account for the noise in estimating $H$, shows how state-of-the-art pruning methods can be adapted efficiently, and delivers strong gains on both LLMs and ViTs.
>
> ## Questions
>
> We thank the reviewer for taking the time to provide these additional questions. We believe we have already addressed the second one in the first part of our response, and we address below the first one:
>
> A single SOCP is tractable, but CAP requires solving many of them. With a block-diagonal approximation of $H$, each block is pruned independently: if a block has $B$ weights, CAP solves $B$ SOCPs to choose the first weight, then $B-1$ for the second, for a total of $K\frac{B(B+1)}{2}$ SOCPs per layer.
>
> We formulated this SOCP and solved it with Gurobi. Although one SOCP takes only 13s on average, the total cost is prohibitive: for the first layer of DeiT-Tiny, we estimate about 137 hours just for the first CAP iteration.
>
> For each layer of DeiT-Tiny, we sampled 50 weights from the $KB$ candidates considered in the first CAP iteration and solved the corresponding SOCPs. We then compared the exact objective in Equation 8 for our approximate solution and at the exact SOCP solution.
>
> |Layer| Obj. improvement (%)|
> |-|-:|
> |0.attn.proj|1.67±0.18|
> |0.attn.qkv|1.88±0.29|
> |0.mlp.fc1|1.06±0.09|
> |0.mlp.fc2|0.85±0.08|
>
> We report only the first layers because of space constraints. The exact SOCP improves the objective by at most 2.02\% on average, but is roughly $350{,}000\times$ slower, suggesting that the small gain does not justify the cost.

---

> > ### Author Rebuttal · Reviewer_xXAq · 2026-04-08
> >
> > My concerns have been adequately addressed.

---

### Official Review · Reviewer_sGwB · 2026-03-12

**Soundness:** 2
**Presentation:** 3
**Significance:** 2
**Originality:** 3
**Overall Recommendation:** 4
**Confidence:** 3

**Summary:**

In previous work of one-shot pruning, the update directions for parameters are often obtained by solving a positive semi-definite quadratic optimization problem involving the Fisher matrix for vision tasks or the second-moment matrix for language tasks. Since these matrices are computationally infeasible to compute exactly, they are replaced by finite-sample approximations. The paper focuses on uncertainty caused by such approximations and proposes a robust optimization framework, called RobOP, that is intended to mitigate such uncertainty in a unified way for both Fisher and second-moment cases. Experimental results on vision and language tasks show accuracy improvements by the robust optimization.

**Compliance With Llm Reviewing Policy:**

Affirmed.

**Final Justification:**

It is still unclear whether the uncertainty in pruning is truly the cause of its accuracy degradation, which forms the motivation of this paper. Nevertheless, since the proposed framework is technically solid and experimental results show its effectiveness, I will keep the positive score.

**Key Questions For Authors:**

- Could you provide some direct evidence for the assumption that the uncertainty arising in one-shot pruning is the main cause of performance degradation for the baseline methods?
- Could you clarify whether the moderate gains justify the added methodological complexity?

**Limitations:**

Limitations are not discussed well in the paper.

**Strengths And Weaknesses:**

**Strengths**

- The paper is overall well written and easy to follow.
- The paper addresses uncertainty in approximated quadratic forms used for one-shot pruning, which seems meaningful at first glance.
- It provides a unified framework for both Fisher and second-moment settings. The derivation based on robust optimization appears technically solid novel and for one-shot pruning.
- Experiments are conducted with standard models and datasets, and the proposed method improves accuracy of one-shot pruning in many cases.

**Weaknesses**

- It is not convincingly justified whether or not the uncertainty arising in one-shot pruning is the main cause of performance degradation for the baseline methods. Indeed, the empirical improvements by the robust optimization are overall moderate relative to the accuracy degradation for higher sparsity settings.
- The variance in experimental results is largely similar between the proposed method and the baselines, which raises a question on the claim that uncertainty in one-shot pruning is effectively mitigated by the proposed framework.

---

> ### Author Rebuttal · Authors · 2026-03-31
>
> We thank the reviewer for their careful reading of the paper and comments.
>
> ## RobOP focuses on estimation noise from $H$
>
> We agree with the reviewer and do not claim that uncertainty in $H$ is the sole source of performance degradation after pruning. Other factors also affect pruning quality, including the heuristics used by pruning methods and the reliance on loss proxies to estimate the impact of pruning on model utility. Our point is more specific: the use of a noisy estimate of $H$ is an important source of degradation in current state-of-the-art pruning methods.
>
> We therefore argue that explicitly accounting for this uncertainty through a robust formulation can substantially improve pruning performance. The gains from RobOP are especially pronounced at high sparsity levels, reaching up to 10 accuracy points at 80\% sparsity for DeiT-Base and 10 perplexity points at 80\% sparsity for Llama-3.1-8B. We also observe meaningful improvements at lower sparsity levels, particularly for ViTs, including a 2.5-point accuracy gain for DeiT-Base at 60\% sparsity.
>
> Finally, the additional experiments conducted in response to Reviewer 4AHW further support our claim that RobOP is particularly effective when uncertainty in $H$ is high, for example when only a small number of samples $N$ is available. In particular, RobOP-CAP improves the accuracy of pruned DeiT-Small by nearly 32 points when only 512 samples are used, while RobOP-ALPS reduces WikiText2 perplexity for pruned Llama-3.1-8B by 33 points when using only 8 sequences of 2048 tokens.
>
> ## Standard errors of RobOP and the baselines
>
> We agree with the reviewer that RobOP does not appear to reduce variability relative to the baseline, and we observe similar standard errors for both our method and the baseline. This is not unexpected, since RobOP reformulates the pruning problem but still relies on existing heuristic pruning methods such as CAP and ALPS to solve it. We therefore believe that the small variations primarily reflect the solution procedures rather than RobOP itself. Given the consistent performance improvements achieved by RobOP, we do not believe this weakens our conclusion that RobOP effectively addresses uncertainty in one-shot pruning.
>
> ## Questions
>
> We thank the reviewer for taking the time to provide these additional questions. We believe we have addressed the first one already in the first part of our response, and we address below the second one.
>
> 2. We believe that the use of RobOP is justified both by its ease of use and by the empirical results.
>
> First, RobOP adds only minimal methodological complexity to the baseline algorithms. The robust formulations require only a simple modification of the original objectives, namely the addition of a convex quadratic regularization term. No further change to the baseline methods is needed. In addition, Table 3 shows that this term can be computed efficiently and introduces little or no computational overhead.
>
> Second, the empirical gains are consistent and, in some cases, very large, reaching up to 10 accuracy points for ViTs and 10 perplexity points for LLMs. As mentioned in the first part of our answer, these gains are even larger in the high-estimation-noise regime, when only a small number of samples is available to estimate $H$.
>
> Finally, RobOP provides a principled interpretation of the parameter $\gamma$, controlling the size of the uncertainty set, which motivates tuning it. This is in clear contrast with the baseline methods, which introduce a regularization term primarily to ensure $H$ is invertible and do not tune $\gamma$. Figures 1 and 2 show that performance can deteriorate sharply when $\gamma$ is misspecified. Therefore, even if the baseline may occasionally perform well with a fixed value of $\gamma$, it can also perform very poorly when that default choice is inappropriate and corresponds to a mis-specified uncertainty set size. This makes our framework more reliable in practice.

---

> > ### Author Rebuttal · Reviewer_sGwB · 2026-04-02
> >
> > Thank you for the detailed rebuttal. However, the first question is still not resolved, since the accuracy improvement by the proposed method does not provide the direct evidence as I expected. It should be provided in a way the proposed method not involved.

---

> > > ### Author Response · Authors · 2026-04-06
> > >
> > > We thank the reviewer again for the detailed feedback. We apologize for not addressing Question 1 in sufficient detail in our previous response, and we provide a more complete clarification here.
> > >
> > > ## Evidence on ViTs.
> > >
> > > First, for ViTs, the experiments show large and consistent improvements, including gains of up to 10 accuracy points for DeiT-Base at 80\% sparsity. We agree that this corresponds to a high sparsity regime. However, the gains are not limited to such high sparsity levels: even at 50\% sparsity, we observe improvements of up to 1 accuracy point, and at 60\% sparsity, improvements of up to 3 accuracy points. These results support our central claim that uncertainty in one-shot pruning is an important source of performance degradation, and that our framework can effectively mitigate it.
> > >
> > > ## Evidence on LLMs.
> > >
> > > For LLMs, following standard practice in the literature, pruning is performed using a calibration set derived from C4, and our robust formulation is likewise applied using this same calibration data. In this setting, we observe consistent and significant improvements in C4 perplexity, again indicating that uncertainty in one-shot pruning is an important factor in performance loss and that our framework is able to account for it effectively.
> > >
> > > We then evaluate the resulting pruned models on additional downstream datasets that are unseen during pruning, including WikiText2 and PTB as well as several classification benchmarks. Despite using only C4 for pruning, the gains seem to transfer across these downstream tasks: we observe perplexity improvements of up to 10 points on WikiText2 and PTB, and gains of up to 0.7 points in mean classification accuracy.
> > >
> > > ## Additional evidence from varying $N$.
> > >
> > > Our latest experiments varying $N$ provide further experimental evidence. In particular, when $N$ is small and estimation noise is high, the benefits of our method become even more pronounced.
> > >
> > > For DeiT-Small at 70\% sparsity, we compare RobOP-CAP and CAP across values of $N$:
> > >
> > > **DeiT-Small test accuracy on ImageNet-1K under different robust formulations after RobOP-CAP pruning.**
> > >
> > > |$N$|512|1024|2048|4096|
> > > |-|-:|-:|-:|-:|
> > > |Baseline|18.48±1.71|39.13±0.64|47.20±0.17|53.77±0.88|
> > > |$\lVert\Delta\rVert_F\leq\gamma$ or $\lVert\Delta_{:,i}\rVert\leq\sqrt{\gamma}$|**50.44±0.75**|**55.29±0.12**|**59.07±0.19**|**60.04±0.32**|
> > > |$\|\Delta_{:,i}\|\leq(\sqrt{\gamma}/N^{1/4})\|\nabla \ell_i(\bar{w})\|_2$ or $\|\Delta\|_F\leq\gamma \mathrm{Tr}(H)/\sqrt{N}$|33.90±0.52|45.87±0.53|54.21±0.25|57.91±0.56|
> > > |$\|\mu_i\|\leq\gamma\sqrt{\frac{\xi_i\mathrm{Tr}(H)}{N}}$|36.59±1.86|46.52±1.32|53.07±0.52|56.80±1.17|
> > >
> > > For Llama-3.1-8B at 70\% sparsity, we compare RobOP-ALPS and ALPS:
> > >
> > > **WikiText2 perplexity of Llama-3.1-8B under different robust formulations after RobOP-ALPS pruning. Here, $S$ denotes the sequence length, set to 2048.**
> > >
> > > |$N$|8$S$|32$S$|128$S$|
> > > |-|-:|-:|-:|
> > > |Baseline|100.28$\pm$8.32|38.51$\pm$2.22|30.16$\pm$1.05|
> > > |$\|\Delta\|_F\leq\gamma$|78.82$\pm$7.27|**36.15$\pm$1.80**|29.93$\pm$1.05|
> > > |$\|\Delta\|_F\leq\gamma\frac{\mathrm{Tr}(XX^\top)}{\sqrt{N}}$|**66.80$\pm$2.54**|38.65$\pm$2.47|29.64$\pm$0.82|
> > > |$\|\mu_i\|\leq\gamma\sqrt{\frac{\xi_i\mathrm{Tr}(XX^\top)}{N}}$|75.78$\pm$5.53|38.82$\pm$2.88|**29.53$\pm$0.79**|
> > >
> > > At 70\% sparsity, for example, we observe a gain of 32 accuracy points for DeiT-Small, as well as improvements of up to 1.9 points in mean accuracy and a 62\% reduction in PTB perplexity for Llama-3.1-8B.
> > >
> > > We agree with the reviewer that 70\% sparsity is still a relatively high pruning level, and we are currently running additional experiments at lower sparsity levels. Given the magnitude of the gains at 70\%, we also expect substantial improvements at lower sparsity when $N$ is small and the noise level is high. We will include these additional results in the revised version.
> > >
> > > ## Additional evidence from the effect of $\gamma$.
> > >
> > > Finally, we believe that the visualization of performance as a function of $\gamma$ provides further evidence that uncertainty arising in one-shot pruning is indeed a factor of performance degradation for the baseline methods. Across all settings, we consistently observe a U-shaped curve (Figures 1--4), which is consistent with our framework: when $\gamma$ is too small, the method is insufficiently robust to estimation noise; when $\gamma$ is too large, the formulation becomes overly conservative and introduces more robustness than needed.
> > >
> > > We thank the reviewer again for their active participation and detailed comments. We sincerely hope that we have been able to address and clarify the questions raised.

---

### Official Review · Reviewer_3Bb9 · 2026-03-13

**Soundness:** 2
**Presentation:** 2
**Significance:** 3
**Originality:** 3
**Overall Recommendation:** 4
**Confidence:** 4

**Summary:**

This paper introduces RobOP, a robust optimization framework for one-shot pruning of large vision and language models. The authors provide a theoretical justification for regularization terms in popular pruning methods by framing them through robust optimization to handle uncertainty in Hessian estimates. While the core idea is strong and novel, the current draft needs work regarding experimental transparency, incomplete literature comparisons, and the absence of a limitations section.

**Compliance With Llm Reviewing Policy:**

Affirmed.

**Final Justification:**

My concerns are resolved.

**Key Questions For Authors:**

Please refer to the previous section.

**Limitations:**

No. The paper lacks a dedicated limitations section. The authors should add a "Limitations and Future Work" section to discuss the current scope boundaries (e.g., focusing only on unstructured pruning), the computational overhead of tuning complex uncertainty sets, and how static bounds might behave at high sparsity levels.

**Strengths And Weaknesses:**

**Strengths**
S1: The paper presents a novel framework (RobOP) that theoretically justifies the ad-hoc regularization terms often used in existing one-shot pruning literature.
S2: The framework is modular and flexible.
S3: The empirical results show notable performance gains on both Vision Transformers and LLMs.
S4: The authors clearly position their contribution by focusing specifically on uncertainty in pruning criteria from noisy Hessian estimates, distinguishing this work from prior research on adversarial robustness.

**Weaknesses**
W1: The paper isn't transparent in the experimental setup. Details about calibration data size, sampling methods, hyperparameter tuning costs, and the exact algorithmic integration for RobOP-CAP and RobOP-ALPS seem to be missing. This makes reproducibility difficult.
W2: The related work section could be broader. It omits recent and potentially relevant pruning methods for LLMs, like SNOWS.
W3: There is no dedicated discussion of limitations or future work.
W4: The writing has readability issues. Undefined variables upon first use, incomplete sentences, and dense paragraphs make it difficult to follow the theoretical derivations.
W5: The authors miss the opportunity to connect their technical achievements to broader real-world applications or economic benefits, relying instead on a generic impact statement.

---

> ### Author Rebuttal · Authors · 2026-03-31
>
> We thank the reviewer for their careful reading of the paper and comments.
>
> ## Details on the experimental setup
>
> We thank the reviewer for this comment. We provide several of these details in the paper, but we agree that some of them should be made more explicit for clarity and reproducibility.
>
> Section 4.1 specifies the calibration set sizes $N$ (128 sequences of 2048 tokens for RobOP-ALPS and 4096 samples for RobOP-CAP). Section 4.2 describes the hyperparameter tuning procedure, namely a grid search over $\gamma$, and Section 4.5 reports pruning times for RobOP and the baselines in Table 3.
> For the LLMs, the calibration set is selected uniformly at random. This is consistent with prior work such as ALPS and SparseGPT. For the ViTs, a stratified split is performed. We will add these details in the paper.
>
> Regarding the algorithmic integration, RobOP is obtained through a simple modification of the original non-robust objectives: the robust formulations are obtained by adding a convex quadratic regularization term. As a result, RobOP only changes the estimate of $H$ with this regularization term, without altering the rest of the pruning algorithms. We mention this briefly in the introduction (lines 100--104), but we agree that this should be stated more clearly. We will add a short paragraph after the derivation of the robust formulations to make this integration explicit.
>
> Finally, we will release the code upon acceptance to support reproducibility.
>
> ## Addition of SNOWS to the Related Work section
>
> We thank the reviewer for pointing us to this valuable reference, which will help improve the paper. We agree that SNOWS is a state-of-the-art method for pruning ViTs and should be cited, and we will add a reference and discussion in the Related Work section. We note that SNOWS refines an existing pruning mask rather than learning one from scratch, which distinguishes it from methods such as RobOP and CAP. In this sense, we believe CAP remains a leading from-scratch pruning method for ViTs.
>
> ## Limitations of RobOP
>
> We agree with the reviewer that it is important to discuss the limitations of our method and outline directions for future work. We will add a dedicated section on these points in the revised version.
>
> ## Clarity of Writing
>
> We thank the reviewer for this comment, which will also help improve the clarity of the paper. We carefully re-read the paper, but we were not able to identify the undefined variables or incomplete sentences mentioned in the review. We would appreciate it if the reviewer could point us to the relevant lines so that we can address these issues in the revised version.
>
> ## Implications of RobOP for Real-World Applications
>
> We thank the reviewer for this feedback. We agree that the practical relevance of pruning, and of RobOP in particular, should be emphasized more clearly. In the revised version, we will expand the introduction to better highlight the economic and practical implications of pruning, including the high cost of serving billion-parameter LLMs and the importance of efficient edge inference for vision models.

---

> > ### Author Rebuttal · Reviewer_3Bb9 · 2026-04-03
> >
> > My concerns are resolved.

---

> > > ### Author Response · Authors · 2026-04-06
> > >
> > > We thank the reviewer for their recognition of our work, and we will carefully incorporate their suggestions in the revised version.

---

### Official Review · Reviewer_4AHW · 2026-03-13

**Soundness:** 2
**Presentation:** 3
**Significance:** 2
**Originality:** 2
**Overall Recommendation:** 2
**Confidence:** 3

**Summary:**

This paper presents RobOP, an optimization framework for one-shot pruning that accounts for uncertainty in Hessian/Fisher estimates caused by limited calibration data, approximation error, and stale curvature during pruning. RobOP derives robust counterparts that add structured PSD regularizers to the nominal Hessian.

**Compliance With Llm Reviewing Policy:**

Affirmed.

**Final Justification:**

The rebuttal improved clarity but did not fully resolve my concerns.

**Key Questions For Authors:**

The paper’s main motivation is uncertainty from small calibration sets. Can you provide results varying the calibration set size for both ViTs and LLMs, and show whether RobOP’s advantage grows as N decreases?

Can you report the actual gamma values selected for the main results in Tables 1 and 2? Are these values stable across seeds, sparsity levels, and model sizes, or is the method highly sensitive?

Figure 1 shows that performance improves over the baseline only in a specific gamma region, then falls off a cliff. The paper should discuss whether the proposed method can be fragile if the uncertainty size is mis-specified.

**Limitations:**

N/A.

**Strengths And Weaknesses:**

### Strengths:

This paper takes multiple sources of Hessian uncertainties (finite calibration sets, noise into account and proposes the RobOP to significantly alleviate those real and under-discussed issues in one-shot pruning.

The framework is flexible and can be instantiated across two important settings: Fisher-based pruning for vision models and layer-wise reconstruction for LLMs.

### Weakness

The strongest-performing variant in several experiments is just adding gamma to H, which prior pruning methods already do as dampening for numerical stability, as the authors themselves acknowledge in Section 3.3 on Page 6. The paper’s novelty is to demonstrate it from the view of robust optimization and the argument for tuning gamma to a more complex format. It is not a new or the final algebraic form in the best case.

A major part of motivation is that small calibration sets make Hessian estimates noisy. The experiments fix the calibration set size and never vary it. This is a serious gap because it leaves the main causal mechanism unverified. If RobOP is supposed to help under sample uncertainty, it should show how performance changes as the number of calibration samples changes.

Although the paper claims to address Hessian uncertainty arising from algorithmic iterative pruning, the proposed framework does not directly model these sources in a clearly distinguishable manner. It remains unclear to what extent RobOP truly mitigates Hessian uncertainty by sequential pruning beyond handling finite-sample estimation noise.

---

> ### Author Rebuttal · Authors · 2026-03-31
>
> We thank the reviewer for their careful reading of the paper and comments.
>
> ## Novelty of our RobOP Framework
>
> We agree that one contribution of RobOP is to reinterpret existing regularization terms through robust optimization. However, this is not the paper’s only contribution.
>
> First, RobOP introduces several robust formulations for both the layer-wise reconstruction error and the Fisher loss, and shows that robustness can be incorporated through simple convex quadratic regularizers that are efficient to compute. This yields novel regularization terms beyond $\gamma I$. While $\gamma I$ is competitive for ViTs, our experiments show it is not the best choice for LLMs. We believe the systematic comparison of uncertainty sets is itself valuable.
>
> Second, RobOP shows that $\gamma$ should be viewed as a meaningful robustness parameter, not just a numerical trick to make $H$ invertible. Unlike prior baselines that fix $\gamma$ to a default value, our results show that poor choices of $\gamma$ can substantially hurt performance (Figures 1 and 2), making tuning important in practice.
>
> Finally, RobOP yields strong empirical gains over the baseline, with up to 10 points higher accuracy for ViTs and 10 points lower perplexity for LLMs.
>
> ## Effect of calibration set size
>
> We agree that varying the calibration size $N$ is an important ablation, especially since RobOP should be most helpful when $N$ is small and the Hessian estimate is more uncertain.
>
> For DeiT-Small at 70\% sparsity, we compare RobOP-CAP and CAP across values of $N$:
>
> **DeiT-Small test accuracy on ImageNet-1K under different robust formulations after RobOP-CAP pruning.**
>
> |$N$|512|1024|2048|4096|
> |-|-:|-:|-:|-:|
> |Baseline|18.48±1.71|39.13±0.64|47.20±0.17|53.77±0.88|
> |$\lVert\Delta\rVert_F\leq\gamma$ or $\lVert\Delta_{:,i}\rVert\leq\sqrt{\gamma}$|**50.44±0.75**|**55.29±0.12**|**59.07±0.19**|**60.04±0.32**|
> |$\|\Delta_{:,i}\|\leq(\sqrt{\gamma}/N^{1/4})\|\nabla \ell_i(\bar{w})\|_2$ or $\|\Delta\|_F\leq\gamma \mathrm{Tr}(H)/\sqrt{N}$|33.90±0.52|45.87±0.53|54.21±0.25|57.91±0.56|
> |$\|\mu_i\|\leq\gamma\sqrt{\frac{\xi_i\mathrm{Tr}(H)}{N}}$|36.59±1.86|46.52±1.32|53.07±0.52|56.80±1.17|
>
> For Llama-3.1-8B at 70\% sparsity, we compare RobOP-ALPS and ALPS:
>
> **WikiText2 perplexity of Llama-3.1-8B under different robust formulations after RobOP-ALPS pruning. Here, $S$ denotes the sequence length, set to 2048.**
>
> |$N$|8$S$|32$S$|128$S$|
> |-|-:|-:|-:|
> |Baseline|100.28$\pm$8.32|38.51$\pm$2.22|30.16$\pm$1.05|
> |$\|\Delta\|_F\leq\gamma$|78.82$\pm$7.27|**36.15$\pm$1.80**|29.93$\pm$1.05|
> |$\|\Delta\|_F\leq\gamma\frac{\mathrm{Tr}(XX^\top)}{\sqrt{N}}$|**66.80$\pm$2.54**|38.65$\pm$2.47|29.64$\pm$0.82|
> |$\|\mu_i\|\leq\gamma\sqrt{\frac{\xi_i\mathrm{Tr}(XX^\top)}{N}}$|75.78$\pm$5.53|38.82$\pm$2.88|**29.53$\pm$0.79**|
>
> The advantage of RobOP grows as $N$ decreases. For example, with 512 samples, RobOP-CAP improves DeiT-Small by nearly 32 accuracy points, and with only 8 sequences of 2048 tokens, RobOP-ALPS improves Llama-3.1-8B by 33 perplexity points on WikiText2. The same trend holds on the other perplexity benchmarks and downstream tasks, including up to 1.9 additional mean accuracy points and a 62\% reduction in PTB perplexity with 8 sequences of 2048 tokens. We will add these results to the paper.
>
> ## Hessian uncertainty from algorithmic updates
>
> In the Fisher-loss setting, iterative pruning in CAP changes the model weights and therefore perturbs both the Hessian and the per-sample gradients used to estimate it. RobOP models this uncertainty in two ways: a Frobenius-norm budget on $\Delta$ (Eq. (5)), which captures global uncertainty in $H$, and a budget on each $\Delta_i$ (Eq. (6)), which captures uncertainty in the $i$-th per-sample gradient. The first is a more generic uncertainty set for $H$, while the second is specifically motivated by pruning-induced noise in the per-sample gradients.
>
> ## Questions
>
> We thank the reviewer for taking the time to provide these additional questions. We believe we have addressed question 1 in the first part of our response, and we address the remaining points below:
>
> 2. We computed the optimal $\gamma$ values used in Tables 1 and 2. For the Fisher loss, they are fairly stable across sparsity levels and architectures and depend mainly on the uncertainty set. For the layer-wise reconstruction loss, they vary more across sparsities and models, which further motivates careful tuning. We cannot include these tables here due to space limits, but we will add them to the appendix.
>
> 3. We agree that Figure 1 highlights an important practical point: arbitrary choices of $\gamma$ can strongly affect pruning performance when the uncertainty set size is mis-specified. This also reveals a weakness of the baseline, which does not tune $\gamma$ and can perform poorly when the default value is inappropriate. In contrast, RobOP explicitly treats $\gamma$ as a tunable robustness parameter, making it more reliable in practice.

---

> > ### Author Rebuttal · Reviewer_4AHW · 2026-04-04
> >
> > Thank you for the rebuttal. After going through other reviews, I think some of my concerns remain unresolved.
> >
> > 1. For gamma, the authors acknowledge that gamma values "vary more across sparsities and models" for the layer-wise reconstruction loss. This actually confirms my concern about sensitivity rather than alleviating it.
> >
> > 2. The fragility issue raised in Q3 is confirmed but not resolved. The rebuttal states that "arbitrary choices of gamma can strongly affect pruning performance," which is precisely the concern I raised. The proposed solution, treating gamma as a tunable hyperparameter, implicitly requires a grid search over gamma before deployment. The pruning times reported in Table 3 do not account for this additional tuning cost, meaning the efficiency claims are overstated. Reframing a sensitivity problem as a design feature does not constitute a resolution.
> >
> > 3. The novelty concern remains. The authors acknowledge that one contribution of RobOP is to reinterpret existing regularization terms through robust optimization. The strongest-performing practical variant in several settings remains the gamma I or gamma H regularizer, which is algebraically equivalent to existing dampening heuristics. The genuinely new robust formulations (e.g., eigenvalue uncertainty sets) don't consistently outperform the simpler variants, making the incremental contribution of the more complex formulations unclear.
> >
> > 4. The causal mechanism for algorithmic-update uncertainty is still unverified. The rebuttal describes two uncertainty sets intended to model pruning-induced perturbations, but provides no experiment that isolates this source of uncertainty from finite-sample estimation noise. Without such ablation, it remains unclear how much of the observed improvement is attributable to each source of uncertainty.

---

> > > ### Author Response · Authors · 2026-04-05
> > >
> > > We thank the reviewer again for the careful reading of our paper and for the thoughtful follow-up comments.
> > >
> > > ## 1. Stability of $\gamma$.
> > >
> > > We would like to emphasize that the optimal $\gamma$ appears to be relatively stable in the ViT setting. For the layer-wise reconstruction loss, the optimal $\gamma$ is less stable, but we view this as an interesting empirical finding rather than a limitation of the framework.
> > >
> > > For the layer-wise and Fisher based settings, we also examined the effect of $\gamma$ on performance. Across our experiments, we consistently observe a U-shaped curve (Figures 1--4), which aligns with our framework: when $\gamma$ is too small, the method is insufficiently robust to estimation noise; when $\gamma$ is too large, the formulation becomes overly conservative and introduces more robustness than needed.
> > >
> > > ## 2. Sensitivity to $\gamma$ and tuning cost.
> > >
> > > We agree that $\gamma$ has a strong impact on performance. In our view, this is especially problematic for the baseline, which is agnostic to estimation noise and therefore fixes $\gamma$ to a constant value. By contrast, our framework explicitly exposes and addresses this issue, which we consider a strength rather than a weakness.
> > >
> > > We also agree that, although we show that the robust formulation can be solved with negligible overhead for a fixed $\gamma$ (Table 3), tuning $\gamma$ does introduce an additional computational cost. We believe this cost is justified in the pruning setting, where pruning is typically performed once and the resulting model is then reused across multiple downstream applications. More broadly, our paper is, to the best of our knowledge, the first to identify estimation noise as an important issue in one-shot pruning, and to propose a robust optimization framework with multiple uncertainty sets and tractable robust formulations, together with simple adaptations of existing baselines that make them robust in practice. We agree that it would be desirable to determine an optimal $\gamma$ without tuning, but we believe this is a challenging problem and leave it for future work.
> > >
> > > ## 3. Novelty beyond reinterpretation of regularizers.
> > >
> > > While we agree that reinterpreting regularizers from prior work through the lens of robustness is a contribution of our paper, we would like to emphasize that it is not the only one.
> > >
> > > Our work is the first to identify estimation noise in one-shot pruning as an important problem and to propose a principled robust optimization framework to address it. We introduce several theoretically motivated uncertainty sets for two widely used state-of-the-art one-shot pruning objectives, and derive the corresponding robust formulations (Theorems 3.1, 3.3, 3.8, and Corollary 3.5). For the layer-wise reconstruction objective, we further adapt concentration bounds from covariance estimation to the second-moment setting (Theorems 3.4, 3.6, and Lemma 3.7). On the algorithmic side, we show that our framework can be incorporated into existing state-of-the-art methods such as ALPS and CAP in a simple manner, yielding substantial empirical gains. We also thank the reviewer for encouraging the evaluation in the smaller-$N$ regime, where we observe improvements of up to 32 accuracy points for DeiT-Small and reductions of up to 62\% in PTB perplexity and 33\% in WikiText2 perplexity for Llama-3.1-8B.
> > >
> > > ## 4. Sources of uncertainty.
> > >
> > > We thank the reviewer for raising this point. As noted in the paper, we consider two sources of uncertainty: finite-sample estimation noise and pruning-induced perturbations. The latter applies only to the Fisher loss, since pruning and weight updates affect both $H$ and the per-sample gradients. In contrast, for the layer-wise reconstruction objective, finite-sample estimation noise is the main source of uncertainty, because pruning does not affect the input and therefore does not change $XX^\top$. Moreover, the input to each layer already reflects the pruning decisions made in the preceding layers.
> > >
> > > The ablation over $N$ provides useful evidence about the impact of these two sources of uncertainty in the Fisher loss. In particular, the performance loss of the baseline starts to plateau at $N=2048$ and $N=4096$, suggesting that the finite-sample estimation noise has become relatively small and that the pruning-induced perturbations are the dominant source of error. By contrast, when $N$ is smaller ($N=512$ and $N=1024$), the finite-sample noise is much more important, leading to a substantial degradation in the non-robust formulation and larger gains from our robust approach.
> > >
> > > We would like to thank the reviewer again for their active participation and detailed comments. We sincerely hope that we have been able to address and clarify the questions raised.

---

### Decision · Program_Chairs · 2026-04-30

**Decision:**

Accept (regular)

**Comment:**

This paper received all positive reviews except one reject. The reject review provides valid concerns and in the end it says, it is OK if we accept the paper. On the positive side, reviewers mention that it is a well-written paper that introduces a novel and modular robust optimization framework (RobOP) to theoretically justify regularization in one-shot pruning, rigorously addressing uncertainty in noisy Hessian-based criteria and demonstrating consistent empirical gains across Vision Transformers and LLMs. Hence, the AC will recommend accepting it. We strongly suggest that authors go through the weaknesses particularly the ones pointed out by the negative review and address them in the final version.